# Auto-GDA: Automatic Domain Adaptation for Grounding Verification in Retrieval-augmented Generation

**Tobias Leemann**[*]
University of Tübingen
tobias.leemann@uni-tuebingen.de

**Periklis Petridis**[*]
MIT
periklis@mit.edu

**Giuseppe Vietri**
AWS AI Labs
vietrigv@amazon.com

**Dionysis Manousakas**
AWS AI Labs
dionman@amazon.com

**Aaron Roth**
AWS AI Labs
aaronrot@amazon.com

**Sergül Aydöre**
AWS AI Labs
saydore@amazon.com

## Abstract

While retrieval-augmented generation (RAG) has been shown to enhance factuality of large language model (LLM) outputs, LLMs still suffer from hallucination, generating incorrect or irrelevant information. A common detection strategy involves prompting the LLM again to assess whether its response is grounded in the retrieved evidence, but this approach is costly. Alternatively, lightweight natural language inference (NLI) models for efficient grounding verification can be used at inference time. While existing pre-trained NLI models offer potential solutions, their performance remains subpar compared to larger models on realistic RAG inputs. RAG inputs are more complex than most datasets used for training NLI models and have characteristics specific to the underlying knowledge base, requiring adaptation of the NLI models to a specific target domain. Additionally, the lack of labeled instances in the target domain makes supervised domain adaptation, e.g., through fine-tuning, infeasible. To address these challenges, we introduce Automatic Generative Domain Adaptation (Auto-GDA). Our framework enables unsupervised domain adaptation through synthetic data generation. Unlike previous methods that rely on handcrafted filtering and augmentation strategies, Auto-GDA employs an iterative process to continuously improve the quality of generated samples using weak labels from less efficient teacher models and discrete optimization to select the most promising augmented samples. Experimental results demonstrate the effectiveness of our approach, with models fine-tuned on synthetic data using Auto-GDA surpassing the performance of the teacher model and reaching the performance level of LLMs at 10 % of their computational cost.

## 1 Introduction

Large Language Models (LLMs) are increasingly used in consequential applications. Despite their versatility, LLMs often produce hallucinations, in which the generated information is inaccurate or fabricated and require costly retraining to integrate new knowledge. One promising method to mitigate these problems is retrieval-augmented generation (RAG, Lewis et al., 2020). RAG enhances text generation by adding information from external knowledge sources to the prompt and has been shown to reduce hallucinations in practice (Shuster et al., 2021). Nevertheless, even when modern LLMs are used with RAG, hallucination rates of 15% – 30% (Chen et al., 2023a) or more than one hallucination per 100 output tokens can occur (Niu et al., 2024).

To prevent hallucinated output from being delivered to end-users, natural language inference (NLI) models can be used to verify the grounding of the generated output in the documents retrieved (Chen et al., 2023b; Es et al., 2024; Tang et al., 2024) before the output is relayed to the end-user: the

---

[*]Work done during internship at AWS AI Labs.

generated response must be fully *grounded* in the documents, i.e., it must be logically inferrable from the documents; otherwise, it is considered ungrounded. However, as we need to check the outputs at inference time, we require lightweight NLI models with very low latency. The current landscape of available NLI models for verifying grounding in RAG is illustrated in Figure 1 based on results obtained in our evaluation of correctness and inference time (see Table 3 for full numeric results): Some recent works such as Mini-Check (Tang et al., 2024) have developed lightweight models for NLI, e.g., based on RoBERTa (Liu, 2019). These models have shown good performance on academic benchmarks. However, our results indicate that their performance in verifying grounding for

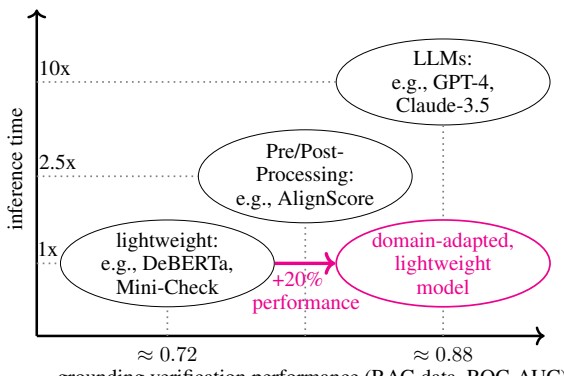

Figure 1: **Landscape of current grounding verification models for RAG.** While LLMs have the best performance, they incur about $10\times$ higher latency than lightweight models. In this work, we are interested in obtaining lightweight models with LLM-level performance for grounding verification through domain adaptation.

realistic RAG inputs lags behind LLMs by about 20% (in ROC-AUC scores). Other recent methods use pre- and post-processesing techniques such as sentence tokenization or LLM prompting to decompose long prompts (Zha et al., 2023; Es et al., 2024) into several chunks or facts. Each of these chunks needs to be processed in a separate forward pass, resulting in high latency as well. While some studies (e.g., Manakul et al., 2023; Tang et al., 2024) have also explored directly using LLMs like GPT-4 for text entailment detection, their latency is about an order of magnitude above the lightweight models. Taken together, these characteristics make it hard to deploy the existing approaches in real-time industry use-cases.

The performance gap observed for realistic RAG inputs with the lightweight models may point to a substantial domain mismatch between the NLI datasets used to train these models and the challenging, real-world data encountered at test time. We observe that inputs of NLI models in RAG are more challenging as they comprise longer segments with multiple statements and contain more subtle ungrounded information as the output is LLM-generated. While these characteristics are common to RAG systems in general, each implementation still has a very individual input distribution: First, inputs may follow a specific format due to the RAG prompt template e.g., *question: <question>. evidence: Passage 1 <evidence1>, Passage 2 <evidence2> ....* Second, the documents are retrieved from knowledge bases from a variety of different domains, which may not be represented in training data. Prior work (Williams et al., 2018) confirms difficulties when NLI models are applied to data from an unseen domain and Hosseini et al. (2024) shows a generalization gap of up to 20%. This suggests that NLI models need to be adapted to their target domain for optimal performance.

Bridging this domain gap poses a significant challenge due to the inherent difficulty of adapting models to unseen domains that is further amplified by the prohibitive costs of obtaining labeled data from the target domain. This prevents supervised domain adaptation, e.g., through fine-tuning on target domain data. To address this issue, we propose *Automatic Generative Domain Adaptation* (Auto-GDA). Our unsupervised domain adaptation framework produces high-quality synthetic data, which is then used to fine-tune a lightweight NLI model, adapting it to a specific domain of RAG inputs. While training data generation by simply prompting LLMs has been repeatedly explored in the literature (e.g., Saad-Falcon et al., 2024; Hosseini et al., 2024), data quality might be further improved through filtering and incorporating background knowledge through label-preserving data augmentation strategies, such as round-trip translation (Chen et al., 2023b). However, specifying good filters and heuristic augmentation strategies require significant manual effort. As data augmentations can further be applied iteratively, the space of potential samples grows exponentially, necessitating efficient search strategies. During this offline training phase, less efficient teacher models can provide additional guidance using weak labels. Auto-GDA offers a unified way to leverage all these available tools. We thus make the following contributions:

1. We formalize the unsupervised domain adaptation problem under the availability of practical tools such as data generators, data augmentation routines, and weak teacher models.

2. We propose Automatic Generative Domain Adaptation (Auto-GDA), a principled framework for unsupervised domain adaptation through synthetic data that can be instantiated with different implementations of generation, augmentation, and weak labeling steps and which automatically selects high-quality samples.

3. We show that our objective corresponds to an *enhanced distribution matching objective* but is highly efficient to optimize. w

4. Our experiments on realistic RAG inputs highlight that our fine-tuned models using Auto-GDA (1) often outperform their weak teacher models (2) perform almost as well as reference models fine-tuned with human-labeled data and (3) reach the level of performance exhibited by LLMs while having almost 10x lower latency. (4) Our method further outperforms more classical training-based unsupervised domain adaptation techniques.[1]

## 2 RELATED WORK

The problem of domain adaptation is concerned with adapting existing models to different domains. We introduce the most closely related approaches in this section and refer the reader to Ramponi & Plank (2020) for further references.

**Synthetic NLI Data.** Related works explore synthetic data generation for NLI models. Hosseini et al. (2024) generate the diverse, cross-domain GNLI (General NLI) dataset synthetically in two steps: first prompting an LLM to generate target domains, then using a prompt-tuned LLM to generate training statements. Tang et al. (2024) generate synthetic training data for their MiniCheck models using document-to-claim generation and claim-to-document generation. We compare to their model in our experimental section and show that it can be further improved through domain adaptation. Saad-Falcon et al. (2024) use synthetic data to specifically improve RAG system evaluation. They generate synthetic in-domain data with a few-shot prompt. However, their method is compared within RAG evaluation frameworks and not tested for NLI performance.

**Synthetic Data for Domain Adaptation in NLP.** While synthetic QA data generation is well-explored (Shakeri et al., 2020; Ushio et al., 2022; Yue et al., 2022; Lee et al., 2023), synthetic data for NLI domain adaptation has received less attention, potentially due to the difficulty of generating realistic and difficult samples. Wang et al. (2023a) propose an iterative synthetic data generation scheme requiring partially labeled data. They generate initial seed data using an LLM prompt that is iteratively refined based on errors from a model trained on human-labeled reference set. Unlike this work, we assume very limited access to labeled data from the target domain.

**Classical Unsupervised Domain Adaptation.** Beyond synthetic data approaches, classical unsupervised domain adaptation (UDA) techniques have also been applied in NLP. These include techniques to align embeddings (Chen et al., 2018; Li et al., 2018; Choudhry et al., 2022; Sun & Saenko, 2016), pre-training approaches (Gururangan et al., 2020; Han & Eisenstein, 2019) or virtual adversarial training (Miyato et al., 2019; Jiang et al., 2020). See Appendix A for details.

We borrow the term "teacher model" from the knowledge distillation literature (Gou et al., 2021; Yang et al., 2020). However, our problem differs from distillation problems because our target dataset is unlabeled. In this paper, we focus on the problem of systematically generating and selecting the most beneficial synthetic samples that can be created through initial generation and iterative augmentation steps. We do so using an efficient objective that can be interpreted as a form of distribution matching.

## 3 PRELIMINARIES

*Domain adaptation* is concerned with adapting an ML model pretrained on a source domain to make predictions on a target domain when the underlying data distributions differ across the two domains. The *unsupervised* domain adaptation problem is further complicated due to the lack of labeled data in the target domain. While features are available, there is no direct information about the correct

---

[1]Code is available at `https://github.com/amazon-science/AutoGDA-Efficient-Grounding-Verification-in-RAG`

class labels for the target domain samples. This poses a significant challenge as the model must learn to adapt to the new distribution without explicit guidance.

## 3.1 UNSUPERVISED DOMAIN ADAPTATION FOR NLI

**Data Domains.** Following the common natural language inference (NLI) setup, we assume data from a *source domain* is available as a set of triples $\mathcal{D}_s = \{(\boldsymbol{e}_n, \boldsymbol{c}_n, y_n)\}_{n=1,\dots,N} \overset{\text{i.i.d.}}{\sim} p_s$ containing evidence $\boldsymbol{e} \in \mathcal{X}$, corresponding claims $\boldsymbol{c} \in \mathcal{X}$ where $\mathcal{X}$ denotes a space of text sequences, and labels $y \in \mathcal{Y}$. This data is used to train an initial model $f : \mathcal{X} \times \mathcal{X} \to \mathcal{Y}$. We use $\mathcal{D}_s$ to denote sets of samples and $p_s$ to denote the data density of the source distribution. Note that in a RAG use case, the evidence $\boldsymbol{e}$ will contain the user prompt as well as the retrieved documents. Additionally, we are provided with a set of $J$ unlabeled samples $\mathcal{D}_t = \{(\boldsymbol{e}_j, \boldsymbol{c}_j)\}_{j=1,\dots,J}$ from the target domain. They are sampled from $p_t$, the data distribution faced at test time (e.g., the realistic RAG inputs). Our goal is to adapt a model pretrained on $p_s$ to perform well on $p_t$. In this work, we are focusing on problems where $J$ is small. This scarcity makes it challenging to accurately estimate the underlying distribution of the target domain, which can hinder the effectiveness of traditional domain adaptation methods that rely on a substantial amount of target data. We study the binary NLI task where $\mathcal{Y} \in \{0, 1\}$. A positive label ($y{=}1$) is only assigned if all information in the claim can be inferred directly from the evidence; claims that are contradictory to the evidence or cannot be inferred from the evidence are considered non-entailed ($y{=}0$).

In this work, we focus on *covariate shift* between the two domains: While the prior $p(\boldsymbol{e}, \boldsymbol{c})$ is subject to change across domains, the true relation between specific features and labels, $p(y|\boldsymbol{e}, \boldsymbol{c})$ is consistent for the source and the target domain. For the NLI task considered here, this assumption is sensible because the entailment relation itself does not change for different domains. Following prior work (Saad-Falcon et al., 2024), we slightly deviate from the fully unsupervised setup by supposing that a very small portion of the target domain can be manually labeled and used as a validation set for hyperparameter tuning only, as is commonly done in NLI literature (Laban et al., 2022; Tang et al., 2022; Zha et al., 2023). We show that our method works with validation sets as small as 30 samples.

**Helper Tools.** We extend this common setup to incorporate three additional tools that are readily available in practice: First, we have powerful generative LLMs that we can use to generate new samples based on the unlabeled examples using techniques such as prompt-tuning (Lester et al., 2021), or few-shot prompting. The *generator $G$* can be formally described as (randomized) function $G : \mathcal{X} \times (\mathcal{X})^F \times \mathcal{Y} \to \mathcal{X}$, meaning that $G$ takes as input a piece of evidence and a set of $F \geq 1$ example claims (e.g., for few-shot prompting) and a desired target label. The generator $G$ is then tasked with producing a new claim sample that reflects the style of the provided claims and has the specified target label. Note that we provide the $F$ claims without a known label, so they can either be entailed or non-entailed w.r.t. $\boldsymbol{e}$. Second, we can use some background knowledge of the task to define some approximately label-preserving *augmentation* strategies to increase diversity, e.g., using paraphrasing models, round-trip translation or synonym replacements (Chen et al., 2023b). This step can be formalized as a mutation function $M : \mathcal{X} \to \mathcal{X}$ which takes a claim as an input and modifies it while trying to preserve its label. The label-preserving characteristics of these strategies are imperfect, i.e., with a small probability the entailment relation will be affected by the augmentation. Finally, we suppose a *teacher model $T : \mathcal{X} \times \mathcal{X} \to [0, 1]$*, which can be applied to the data from the source and the target domain and provides an entailment score. The teacher model performs reliably within the source domain, but only provides a weak estimate of $T(\boldsymbol{e}, \boldsymbol{c})$. The performance of this model may be noisy because the target domain is out-of-domain for this model, and the model may be too inefficient to be deployed in practice. We will use this model to obtain weak estimates of the samples' labels. We now present our framework Auto-GDA, which incorporates the three tools $G, M, T$ named above in a principled algorithm.

## 4 A PRINCIPLED FRAMEWORK FOR UNSUPERVISED DOMAIN ADAPTATION

### 4.1 OUTLINE OF THE FRAMEWORK

In this work, we present Auto-GDA, a framework for *Automatic Generative Domain Adaptation*, that generates synthetic data points that are useful for fine-tuning a pretrained model $f$ for the target

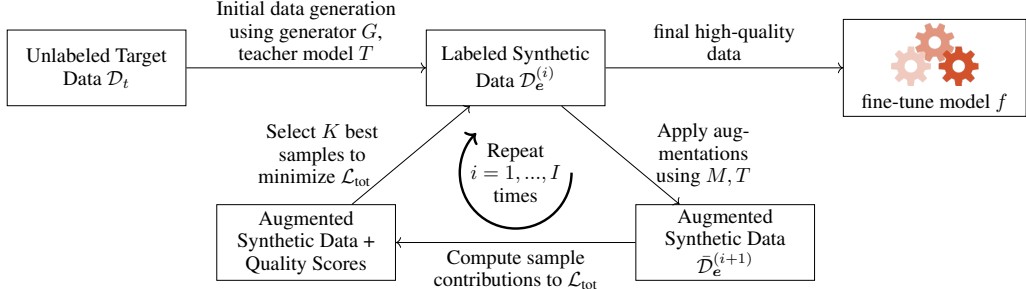

Figure 2: **Overview of Auto-GDA.** We generate initial data using the generator $G$, which are assigned entailment certainty scores using teacher model $T$. The synthetic data is iteratively augmented using $M$, whereas label-preservation is confirmed with $T$ and entailment certainties are updated. We finally select the top-$K$ samples that minimize an objective function $L_{\text{tot}}$. These steps can be applied iteratively until the final data is used to fine-tune the model $f$ for the target domain.

domain. For the data generation process to result in high fine-tuning utility it must meet several criteria: (1) The data must be realistic and non-trivial, (2) must have high diversity, (3) the assigned labels must be of high quality.

Auto-GDA is specifically designed to tackle these three challenges. As RAG outputs stem from LLMs, we also generate realistic initial claim samples for a given evidence using LLMs. We leverage few-shot prompting to transfer patterns in the output to the generated samples. To preserve the *diversity* of the evidence (which contains the relevant documents in the knowledge base), we generate synthetic claims sequentially for each unique piece of evidence $e$ available in the unlabeled target dataset $\mathcal{D}_t$. This has the advantage that the broad diversity of documents in the knowledge base is represented. We propose to apply augmentations on the synthetic data to increase diversity further. As the augmentation strategies are only approximately label-preserving, we have to keep track of increasing label uncertainty to *detect samples with low-quality labels* when several data augmentation steps are applied. We therefore equip each sample with an *entailment certainty score* $r$, an estimate of the probability of the sample having an entailed label ($y=1$) which can be used to remove samples with low-quality labels. Auto-GDA applies these steps iteratively to successively increase data quality. In summary our framework consists of the following steps, which we describe in more detail in the next sections:

1. **Initial Generation.** Generate an initial sample population $\mathcal{D}_e^{(0)} = \{(\hat{c}_k, \hat{y}_k, r_k^{(0)})\}_{k=1}^K$ of claims $\hat{c}$ and labels $\hat{y}$ for the evidence $e$ using the generator $G$. Use the teacher model $T$ to assign initial entailment certainty $r^{(0)}$ scores to each sample of synthetic data. This results in each sample having a hard label $\hat{y}$ and a "soft" confidence score $r^{(0)}$ for the hard label.

2. **Sample Augmentation.** Apply augmentations $M$ on claims in the population $\mathcal{D}_e^{(i)}$ to obtain new claims with the same hard labels. Update their entailment certainties using the teacher model again. Merge mutated samples and samples from previous iteration to form updated population $\bar{\mathcal{D}}_e^{(i+1)} = \{(\hat{c}_l, \hat{y}_l, r_l^{(i+1)})\}_{l=1}^L$ that is of larger size $L \gg K$.

3. **Sample Selection.** Select the subset of samples of size $K$ from $\bar{\mathcal{D}}_e^{(i+1)}$ that minimize our proposed *enhanced distribution matching objective* $\mathcal{L}_{\text{tot}}$ formally introduced in Eqn. (4). The objective includes the unlabeled target samples $\mathcal{D}_t$ and the certainty scores. The selected subset becomes the next generation dataset $\mathcal{D}_e^{(i+1)}$.

4. Repeat steps 2 and 3 for a fixed number of iterations or until objective $\mathcal{L}_{\text{tot}}$ converges.

We illustrate these steps in Figure 2 and will detail out implementation choices for each step below.

## 4.2 GENERATING REALISTIC INITIAL DATA

LLMs have been repeatedly used to generate synthetic data for various domains, including NLI (Saad-Falcon et al., 2024; Hosseini et al., 2024). In this work, we generate initial data using few-shot prompting with the prompts provided in Appendix E.1. The prompt instructs the LLM to generate synthetic claims $\hat{c} = G(e, \text{claim}(\mathcal{D}_{t,e}), \hat{y})$ for the evidence $e$, reflecting the style of ex-

ample claims from $\mathcal{D}_t$ (claim($\mathcal{D}_{t,e}$) denoting claims from target data for the evidence $e$) and target label $\hat{y} \in \{0, 1\}$. For label $\hat{y} = 1$, the LLM is instructed to include only grounded facts, for $\hat{y} = 0$, some ungrounded information should be introduced. We assign labels $\hat{y}$ according to the prompt used, resulting in complete initial generated tuples $(\hat{c}, \hat{y})$. We follow some related works (Puri et al., 2020; Vu et al., 2021), which have suggested generating many samples and only keeping the most confident. To do so, the samples can be equipped with a weak estimate of the label probability using the teacher model, e.g., another LLM or an NLI model with sophisticated pre- and postprocessing. In the binary classification setup, we can compute initial entailment certainties as $r^{(0)} = T(e, \hat{c})$, which can be interpreted as an uncalibrated and potentially noisy estimate of $p(y = 1 | e, \hat{c})$. We explore LLMs for data generation and use state-of-the-art NLI models and also LLMs as teacher models $T$ for providing initial entailment certainties. Adding the entailment certainty scores $r^{(0)}$ to the respective tuples we obtain a set of triples $\mathcal{D}_e^{(0)} = \{(\hat{c}_k, \hat{y}_k, r_k^{(0)})\}_{k=1}^K$ after this step.

## 4.3 Increasing Diversity through Label-Preserving Data Augmentations

In this section, we demonstrate how to augment the initial synthetic dataset (generated using the few-shot prompting strategy) for additional diversity, while maintaining a high degree of label certainty for the augmented synthetic data points. We exploit a certain degree of background knowledge to derive data augmentation strategies (Chen et al., 2023b). For instance, we know that paraphrasing the claims while preserving their semantic meaning should not change their entailment label. However, when iteratively applying paraphrasing operations, we have to account for an increasing probability of accidentally flipping the label.

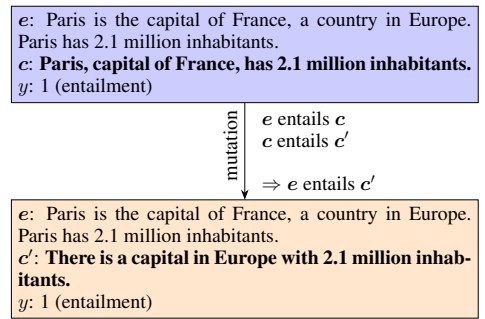

Figure 3: **Intuition for our update rule for entailment certainties:** If a parent claim $c$ is entailed by $e$ and a mutated claim $c\prime$ is entailed by its parent $c$, the mutated claim $c'$ will be entailed by $e$ as well.

**Obtaining High-Quality Entailment Certainties.** We can combine the generative models with discriminative teacher models again to obtain weak estimates $r^{(i)}$ of the entailment certainty of the augmented samples. Instead of directly computing the entailment probability using $T$, we exploit logical invariances, which allow for better estimates depicted in Figure 3: If the original claim is entailed by the evidence, and if the modified claim is entailed by the original claim, the modified claim will also be entailed by the evidence. Suppose we have obtained $\hat{c}' = M(\hat{c})$ as a modification of the synthetic claim $\hat{c}$. As we already have an estimate of the entailment probability for $(e, \hat{c})$, we can reuse it and only need to compute another entailment probability for $(\hat{c}, \hat{c}')$. We argue that computing this entailment probability is easier for the teacher model than directly computing $T(e, \hat{c}')$, as the claim and the modified claim should be semantically and syntactically more similar. Paraphrasing datasets like PAWS (Zhang et al., 2019) are common pretraining datasets, and standard NLI datasets like MNLI (Williams et al., 2018) contain many similar samples due to their construction through edits, so NLI predictions are expected to be more reliable on these pairs. Querying the teacher model on $T(\hat{c}, \hat{c}')$ allows us to use the following update rule for the augmented sample $(e, \hat{c}')$:

$$r^{(i+1)}(e, \hat{c}') = r^{(i)}(e, \hat{c}) \cdot T(\hat{c}, \hat{c}') + (1 - r^{(i)}(e, \hat{c})) \cdot (1 - T(\hat{c}, \hat{c}')). \quad (1)$$

using the entailment certainty $r^{(i)}$ of the original tuple $(e, \hat{c})$ as a base. Some teacher models may be particularly reliable with claim-claim pairs rather than with evidence-claim pairs, so it can be useful to choose a different teacher model for this update than for obtaining initial certainty scores.

**Label Invariant Augmentation Strategies:** In this work, we consider three augmentation strategies that will likely preserve entailment labels (see Appendix C.1 for additional details):

- **Partial Rephrasing with LLMs.** Our first augmentation is an LLM-based rephrasing step. Specifically, we randomly mask 20% of the words of the input sequence by replacing the corresponding words by "_" and ask an LLM (Anthropic's Claude3 Haiku) to impute the gaps while preserving the meaning.

- **Complete Paraphrasing.** We use a T5-based paraphrasing model (Vorobev & Kuznetsov, 2023). We generate paraphrases for the claims enforcing diversity using a constraint that prevents $n$-grams of length greater than 5 from being regenerated.

- **Sentence Deletion.** We chunk the claim into sentences and randomly delete one of them. This should preserve the entailment relation as it only removes information. However, we note that this augmentation may remove some of the context necessary to understand the entire claim.

We generate several augmentations for each sample using these strategies along with an estimate of their entailment probabilities, resulting in an enlarged sample set. Unfortunately, not all of these samples may be of high quality. Therefore, it is crucial to select only the most promising samples.

## 4.4 AUTOMATIC SELECTION OF HIGH-QUALITY SAMPLES

A key component of our work involves automatically selecting the most promising samples. Intuitively, we are interested in finding samples that resemble target data. This includes both having realistic features and correctly assigned labels. The data should also have a high chance of improving the final model. Provided with an augmented dataset $\bar{\mathcal{D}}_e^{(i)} = \{\hat{c}_l, \hat{y}_l, r_l\}_{l=1}^L$ at iteration $i$, we are interested in selecting a subset $\mathcal{Q}_e \subset \bar{\mathcal{D}}_e^{(i)}$ of a smaller size $|\mathcal{Q}_e| = K$ that only contains the most promising samples. We propose the following objective function to assign a loss to a selected subset $\mathcal{Q}_e$ which contains three terms for each selected sample:

$$\mathcal{L}_{tot}(\mathcal{Q}_e, f) = \sum_{\hat{c}_i, \hat{y}_i, r_i \in \mathcal{Q}_e} \left[ \underbrace{d(\hat{c}_i, c_{\min,i})^2}_{\text{distance}} + \lambda_d \underbrace{\text{LDiv}(r_i, \hat{y}_i)}_{\text{label correctness}} - \lambda_u \underbrace{U_f(\hat{c}_i, \hat{y}_i)}_{\text{utility term}} \right], \qquad (2)$$

where $d(x, x') = \|\psi(x) - \psi(x)'\|$ is a distance function over inputs in $\mathcal{X}$ defined via textual embeddings $\psi$, $c_{\min,i} \coloneqq \arg\min_{c' \in \text{claim}(\mathcal{D}_{t,e})} d(c', \hat{c}_i)$ is the closest claim for evidence $e$ from the target dataset, and $\lambda_d, \lambda_u$ are hyperparameters. LDiv : $[0, 1] \times \{0, 1\} \to \mathbb{R}_+$ is a function that penalizes uncertain labels taking the certainty scores $r$ and the hard labels $\hat{y}$ as inputs as plotted in Figure 4. We derive the exact form of the LDiv function as a divergence estimate of the conditional distributions in Appendix B.2. The *distance* term encourages samples to be close to claims from the target data set for the given evidence. The *label correctness* term penalizes samples where the entailment certainties are too far apart from the target labels and is used to discourage selection of samples where the labels are likely to be incorrect. Additionally, we encourage generation of samples where the pretrained model $f$ is not performing well yet by including the cross-entropy loss of the model as a utility term, $U_f = \text{CE}[f(e, \hat{c}), \hat{y}]$ where $\hat{y}$ is the assigned hard label of a synthetic sample.

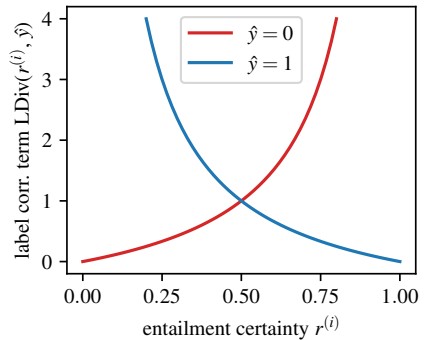

Figure 4: **Modeling the label correctness term in Eqn. 2 as function of $r$.** When the estimated entailment certainty $r$ does not match the assigned hard label $\hat{y}$ this term takes high values, discouraging selection.

**Theoretical Properties.** Notably, Equation (2) can be derived from first principles as an *enhanced distribution matching objective*. By defining parametric distributions $p_{\mathcal{Q},e}(c, y)$ (representing the selected synthetic data for evidence $e$) and $p_{\text{cov},e}(c, y)$ (representing the target distribution for $e$ we aim to imitate) the objective corresponds to the divergence between these distributions minus the expected utility of the synthetic data. Formally,

$$\mathcal{L}_{tot}(\mathcal{Q}_e, f) = D_{KL}\left(p_{\mathcal{Q},e}(c, y) \| p_{\text{cov},e}(c, y)\right) - \mathbb{E}_{(c,y) \sim p_{\mathcal{Q},e}}\left[U_f(c, y)\right]. \qquad (3)$$

We derive a proposition to formalize this connection in Appendix B.

**Optimizing the objective.** Optimizing the objective for a subset $\mathcal{Q}_e$ containing $K$ synthetic samples for evidence $e$ with minimal loss can be done highly efficiently in three steps: (1) Computing each samples' contribution to the sum in $\mathcal{L}_{tot}$, (2) ranking the samples by this contribution, and (3) greedily selecting the top-$K$ subset of samples with the lowest contributions. Pseudocode of our complete framework is provided in Algorithm 1 (Appendix).

| Dataset | RAGTruth | | LFQA-Verif. | SummEdits | Avg., Rank |
| RAG-Task | Summary | QA | QA | Summary | |
| --- | --- | --- | --- | --- | --- |
| **base models** | | | | | |
| FLAN-T5 | 0.734 | 0.708 | 0.655 | 0.700 | 0.699 |
| BART-large | 0.696 | 0.670 | 0.821 | 0.769 | 0.739 |
| DeBERTaV2 | 0.782 | 0.530 | 0.645 | 0.876 | 0.708 |
| **robustness** | | | | | |
| DAPT$_{\text{DeBERTaV2}}$ | 0.746 ± 0.005 | 0.703 ± 0.016 | 0.813 ± 0.094 | 0.837 ± 0.004 | 0.775 |
| SiFT$_{\text{DeBERTaV2}}$ | 0.785 ± 0.008 | 0.566 ± 0.005 | 0.880 ± 0.032 | 0.845 ± 0.003 | 0.769 |
| CORAL$_{\text{DeBERTaV2}}$ | 0.718 ± 0.001 | 0.677 ± 0.001 | 0.822 ± 0.001 | 0.853 ± 0.001 | 0.768 |
| **complex** | | | | | |
| MiniCheck-T5 | 0.754 | 0.640 | 0.741 | 0.791 | 0.732 |
| AlignScore | 0.729 | 0.822 | 0.904 | 0.894 | 0.837 |
| Vectara-2.1 | 0.805 | 0.854 | 0.648 | 0.590 | 0.725 |
| **Auto-GDA** | | | | | |
| Flan-T5 (Auto-GDA) | 0.756 ± 0.004 | 0.783 ± 0.013 | 0.687 ± 0.002 | 0.824 ± 0.010 | 0.762 |
| BART (Auto-GDA) | 0.813 ± 0.009 | 0.867 ± 0.011 | 0.867 ± 0.026 | 0.860 ± 0.010 | 0.852 ③ |
| DeBERTaV2 (Auto-GDA) | 0.837 ± 0.007 | 0.867 ± 0.007 | 0.925 ± 0.009 | 0.883 ± 0.005 | 0.878 ② |
| **LLMs** | | | | | |
| GPT-3.5 | 0.706 | 0.648 | 0.749 | 0.814 | 0.729 |
| GPT-4o-mini | 0.884 | 0.833 | 0.812 | 0.878 | 0.852 ③ |
| GPT-4o | 0.892 | 0.865 | 0.896 | 0.880 | 0.883 ① |

Table 1: **Performance comparison to baselines (ROC scores).** Grouped by off-the-shelf base models trained on standard data, domain-adapted versions of the best base models using DAPT, SIFT, and DeepCORAL, complex state-of-the-art models trained using custom datasets (Vectara, MiniCheck) or using postprocessing (AlignScore), proprietary LLMs, and versions of the base models fine-tuned with Auto-GDA. We highlight the teacher model that was used to assign initial label certainties $r^{(0)}$ in a ⬚ box and make three observations: (1) the Auto-GDA version of the base models always improves over the vanilla versions and the versions trained with SIFT, Deep CORAL, and DAPT, (2) our best-performing model DeBERTaV2 (Auto-GDA) outperforms its teacher model in three out of four cases, and (3) BART and DeBERTa with Auto-GDA reach LLM-level performance.

## 5 EXPERIMENTAL EVALUATION

We run experiments with realistic datasets and baseline models to confirm the efficacy of Auto-GDA.

**Datasets.** We evaluate our approach on three datasets for document-grounded summarization and question answering (QA). We select datasets that include documents, realistic LLM-generated long-form answers, and human labels that can be used for testing. The SummEdits dataset (Laban et al., 2023) contains GPT-3.5-generated and manual summaries of documents from different domains, e.g., judicial, sales emails, podcasts. We further use both the summary and the QA portion of the RAGTruth dataset (Niu et al., 2024). The RAGTruth dataset contains summaries and answers to questions created by LLMs (GPT-3.5/4, Mistral, Llama2). Finally, we use the LFQA-Verification dataset (Chen et al., 2023a), which retrieved documents for questions from the "Explain me Like I am five"-dataset and generated corresponding long-form answers with GPT-3.5 and Alpaca. We selected the datasets to cover characteristics of realistic RAG systems including specific prompt templates (present in RAGTruth, LFQA-Verification) and various domains (present in all datasets, specifically in SummEdits). Details and links to the datasets can be found in Appendix C.2.

**Base models.** As NLI models, we use three pretrained model architectures that are able to handle NLI queries with the longer context required for RAG inputs. We investigate a BART-Large (Lewis et al., 2019) model pretrained only on the MNLI dataset (Williams et al., 2018). This can be considered a lightly pretrained model. Additionally, we study DeBERTa-V2 pretrained with datasets from the TaskSource collection (Sileo, 2024). We additionally study a FLAN-T5-based model (Raffel et al., 2020) pretrained on MNLI. The all models possess context lengths of at least 1024 tokens.

**Baselines.** We use state-of-the-art baselines: AlignScore (Zha et al., 2023) (RoBERTa-based with pre- and postprocessing), MiniCheck (Tang et al., 2024), and Vectara-2.1[2] (both T5-based). These complex state-of-the-art models are trained using custom datasets (Vectara, MiniCheck) or use postprocessing (AlignScore). As a teacher model to assign initial score, we use the best-performing model from the "complex" category, which (unlike LLMs) allow easy access to uncertainty scores

---
[2] https://docs.vectara.com/docs

| Dataset RAG-Task | RAGTruth Summary | QA | LFQA-Verif. QA | SummEdits Summary | Mean (Gap closed) |
|---|---|---|---|---|---|
| non-fine-tuned | 0.782 | 0.530 | 0.645 | 0.876 | 0.708 (0%) |
| +ft. on Few-Shot Data | 0.799 | 0.826 | 0.934 | 0.872 | 0.858 (84%) |
| +ft. on Augmented w/ Random Selection | 0.777 | 0.783 | 0.919 | 0.862 | 0.835 (71%) |
| +ft. on Augmented w/ Objective (Auto-GDA) | 0.837 | 0.867 | 0.925 | 0.883 | 0.878 (96%) |
| fine-tuned on labeled | 0.842 | 0.890 | 0.909 | 0.898 | 0.885 (100%) |

Table 2: **Ablation:** Fine-tuning with synthetic data obtained by few-shot prompting and random selection of augmentations as opposed to using our framework Auto-GDA. We also report performance relative to the hypothetical upper baseline of fine-tuning on labeled target data and observe that we can almost close this domain-adaptation gap (ROC, DeBERTa model, avg. over 5 runs).

and have acceptable performance. We employ `optuna`[3] as a principled way of choosing the remaining hyperparameters $\lambda'_u$, $\lambda_d$, and the other teacher model used to estimate entailment probabilities for augmentations in Eqn. 1. We perform 50 trials per dataset and use the ROC score of a fine-tuned DeBERTaV2 model on the small validation dataset as selection objective. In case limited budget for hyperparameter tuning is available, we recomment setting $\lambda'_u = \lambda_d \in [20, 50]$ which led to stable performance. Auto-GDA is run for two iterations on RAGTruth and one iteration on the other datasets, generating synthetic datasets between $1.3\times$ and $2\times$ the original dataset size. We found no improvements through further increasing dataset size. We also compare against several common UDA methods, including robustness-based approaches. Specifically, we implement DAPT (Gururangan et al., 2020), SiFT (He et al., 2020), and Deep-CORAL (Sun & Saenko, 2016) for further pretraining of the DeBERTa-V2 model.

## 5.1 SYNTHETIC DATA FOR NLI MODEL FINE-TUNING

We present the main results obtained with Auto-GDA in Table 1. Our results show that Auto-GDA is highly effective and improves performance in ROC-AUC scores of all tested models on all datasets.

**Comparison to Teacher Models.** Auto-GDA is highly effective, not only incorporating the knowledge of the stronger teacher model (indicated by box) but often even surpassing it, as the optimization enhances data quality over the teacher in three out of four datasets.

**Comparison to Classical UDA Methods.** Traditional UDA methods (DAPT, SiFT, and Deep-CORAL) did not yield significant improvements in our NLI domain adaptation setting and Auto-GDA consistently outperforms them across all datasets. This also indicates that synthetic data generation is more effective for NLI tasks.

**Comparison to LLMs.** Finally, our fine-tuned models reach performance levels between state-of-the-art LLMs such as GPT-4o and GPT4o-mini while maintaining significantly lower computational requirements. This shows that our approach results in models with superior NLI performance, in particular when compared to the non-fine-tuned or non-LLM baselines.

**Other Teacher Models.** We investigate using LLMs and other teacher models in Table 9 (Appendix) but observe that LLMs do not generally outperform other teacher models, possibly due to unreliable uncertainty scores. However, the table also shows that the DeBERTa model can improve its own performance through self-supervision by an average of 0.15 AUC when applied as the teacher model.

## 5.2 ABLATION INVESTIGATIONS

**Components of the algorithm.** We add the components of our algorithm individually and show how they successively increase performance in Table 2. In all ablations, we keep dataset size and other parameters constant. The biggest gain is achieved by fine-tuning on data created through few-shot prompting. We subsequently add data augmentations without applying our selection criterion, but instead selecting few-shot and augmented samples randomly. We observe that this decreases data quality, highlighting that data augmentation is only beneficial together with our filtering criterion. When we do so and apply data augmentation with our filtering step (corresponding to

---

[3]https://optuna.org/

| Model | RAGTruth | | LFQA-Verification | SummEdits | Inference time (relative) | Performance |
| | Summary | QA | QA | Summary | sec/(50 samples) | AUC-ROC*100 |
|---|---|---|---|---|---|---|
| Vectara | $1.57 \pm 0.02$ | $1.13 \pm 0.03$ | $1.35 \pm 0.03$ | $1.03 \pm 0.01$ | 1.27 (59%) | 72.5 |
| FLAN-T5 | $1.71 \pm 0.07$ | $1.71 \pm 0.07$ | $1.72 \pm 0.07$ | $1.71 \pm 0.07$ | 1.71 (80%) | 69.9 |
| DeBERTaV2 | $2.56 \pm 0.03$ | $1.88 \pm 0.04$ | $2.15 \pm 0.06$ | $1.88 \pm 0.09$ | 2.12 (100%) | 70.8 |
| MiniCheck-T5 | $4.50 \pm 0.20$ | $3.16 \pm 0.06$ | $3.90 \pm 0.14$ | $3.22 \pm 0.10$ | 3.69 (174%) | 73.2 |
| BART-large | $4.33 \pm 0.01$ | $3.62 \pm 0.06$ | $3.95 \pm 0.09$ | $3.76 \pm 0.20$ | 3.92 (184%) | 73.9 |
| AlignScore | $5.88 \pm 0.12$ | $7.55 \pm 0.28$ | $7.55 \pm 0.35$ | $1.81 \pm 0.06$ | 5.70 (269%) | 83.7 |
| GPT-4o | $19.80 \pm 0.51$ | $19.11 \pm 0.44$ | $21.09 \pm 2.97$ | $21.89 \pm 1.26$ | 20.47 (967%) | 88.3 |
| Auto-GDA $_{\text{DebertaV2}}$ | | | Same as DeBERTaV2 | | 2.12 (100%) | 87.8 |

Table 3: **Inference times of the models** on the datasets as well es average performance taken from Table 1. Our DeBERTa model combines LLM-level performance with substantially lower latency.

full Auto-GDA), this increases performance overall with one exception on the LFQA-Verification dataset (note however that performance here is already above the labeled data, so selection based on target data may draw the results toward the labeled data scores as well). As an upper baseline we are interested in the hypothetical performance reachable by fine-tuning on human-labeled samples and include it in Table 2. Considering the difference between the no fine-tuning models and the models fine-tuned on human-labeled data as the *domain adaptation gap*, expressing our results relative to these baselines indicates that we manage to close an impressive 96% of this gap.

## 5.3 INFERENCE EFFICIENCY

Linking to our motivational Figure 1, we study the efficiency of our models in Table 3. We compute NLI scores for 50 random samples from the respective datasets. We observe models in three categories: The most efficient models (Vectara to BART-large) have medium performance on the RAG datasets used in this work indicated by their ROC. On the other hand, models using sophisticated post-processing (AlignScore) perform better, but require about 2.5 times more inference time than our most successful DeBERTa model. Finally, LLMs via APIs require about 10-fold inference time, but result in highest performance. When we compare models trained with our approach, we observe LLM-level performance at about 10% of the inference time. Although our primary focus lies on inference, we report generation times for Auto-GDA in Table 16 (Appendix) for completeness.

## 6 DISUSSION AND CONCLUSION

In this work, we show that synthetic data can successfully tackle the domain generalization gap for NLI models. We present Auto-GDA, an automatic approach for synthetic sample generation and selection overcoming the need for tedious heuristic or manual filtering and augmentation selection. Our results show that we can obtain models that perform on par with most powerful LLMs while having around 90% less inference time using our method. By generating synthetic data, we can provide a more comprehensive and tailored representation, allowing for greater control over the desired features. Our results also confirm the common intuition that generalization is increasingly hard with smaller models (Bhargava et al., 2021). This highlights that domain adaptation is particularly important when low latency at inference time is required, whereas general purpose models can be preferable when quick inference is no hard requirement. Pretrained models as AlignScore (Zha et al., 2023) and MiniCheck (Tang et al., 2024) strike a good balance between inference time and performance when generation and fine-tuning is not possible or too costly.

**Limitations.** In our study we assume that the distribution of evidence samples including the retrieved documents is readily available. In many real-world applications, this may not be the case. To address this, techniques like passage clustering and summarization, as explored in Sarthi et al. (2024), could be employed on the knowledge base to cover the diversity of evidence passages as a surrogate of this distribution. In general, we advise practitioners to obtain the most representative corpus of the target domain for best results. Currently, domain adaptation requires a model for each individual domain. Future work is required to further study if it is possible to adapt efficient models for multiple domains without performance degradation. In addition, an assessment including the entire RAG system with a retrieval component using frameworks such as RAGChecker (Ru et al., 2024) may result in insightful in future work.

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

## A ADDITIONAL RELATED WORK

**Automatic Data Selection.** A related stream of research is concerned with automatically selecting subsets from large datasets for training. For instance, AutoAugment (Cubuk et al., 2019) searches for optimal image augmentations through reinforcement learning, but can be computationally intensive. In contrast, Xie et al. (2023) propose an efficient importance-weighting criterion based on hashed $n$-gram distributions using Kullback Leibler Divergence (KLD). Data selection is linked to works on data valuation, e.g., Wang & Jia (2023), as data valuation scores can be used to select training data, often resulting in improved performance.

**Classical Unsupervised Domain Adaptation.** Beyond synthetic data approaches, classical unsupervised domain adaptation (UDA) techniques have also been applied in NLP. Chen et al. (2018); Li et al. (2018); Choudhry et al. (2022) have explored Domain Adversarial Neural Networks (DANN) (Ganin et al., 2016), which incorporate domain discriminators during pretraining to learn domain-invariant features. He et al. (2020) introduce Scale-invariant-Fine-Tuning (SiFT) which extends the Virtual Adversarial Training (VAT) framework of Miyato et al. (2019) and Jiang et al. (2020) to improve model robustness and generalizability. Techniques like CORAL (Sun & Saenko, 2016) align feature distributions between source and target domains by matching their second-order statistics. Finally, domain-adaptive pretraining (DAPT) and task-adaptive pretraining (TAPT) (Gururangan et al., 2020; Han & Eisenstein, 2019) involve pretraining on target domain text before fine-tuning on labeled source data. Although these methods have shown success in tasks like sentiment analysis and text classification, they have not been comprehensively studied in NLI.

## B THEORETICAL CONSIDERATIONS

### B.1 DERIVING OUR OBJECTIVE FROM A DISTRIBUTION MATCHING CRITERION

In this section, we present an more formal derivation of our objective $\mathcal{L}_{\text{tot}}$ from an *enhanced distribution-matching* objective. We decrease a statistical divergence between a parametric distribution represented by the selected synthetic data samples $p_{\mathcal{Q}}$ (for instance, their Parzen-Window

estimator or MLE estimate of a parametric family) and a target data distribution providing the regions in feature space that we would like to cover, denoted by $p_{\text{cov}}$. We now consider $\boldsymbol{c}$ to be a vector in a continuous vector space. Such a mapping can be realized through stochastic encoders / decoders. As we consider each evidence $\boldsymbol{e}$ independently, we write $p_{\mathcal{Q},e}(\boldsymbol{c}, y)$ as a shorthand for $p_{\mathcal{Q}}(\boldsymbol{c}, y|\boldsymbol{e})$. Additionally, we encourage generation of samples where the pretrained model $f$ is not performing well yet by including the cross-entropy loss of the model as a utility term, $U_f = \text{CE}[f(\boldsymbol{e}, \hat{\boldsymbol{c}}), \hat{y}]$ where $\hat{y}$ is the assigned hard label of a synthetic sample. In summary, we propose optimizing the distribution parameters $\mathcal{Q}$ to minimize the objective $\mathcal{L}_{tot}$,

$$\min_{\mathcal{Q}} D_{KL}\left(p_{\mathcal{Q},e}(\boldsymbol{c}, y) || p_{\text{cov},e}(\boldsymbol{c}, y)\right) - \mathbb{E}_{(\boldsymbol{c},y)\sim p_{\mathcal{Q},e}}\left[U_f(\boldsymbol{c}, y)\right] := \min_{\theta} \mathcal{L}_{tot}(p_{\mathcal{Q},e}, p_{\text{cov},e}, f) \quad (4)$$

where $U_f$ is an additional per-sample utility term that depends on the model $f$. We omit $\boldsymbol{e}$ from the subscript to shorten notation, but still consider a fixed evidence $\boldsymbol{e}$. Since we do not have labels for the target samples, we can't estimate the target distribution term $p_{\text{cov}}(\boldsymbol{x}, y)$ in the distribution objective $\mathcal{L}_{tot}$. However, we can decompose the divergence using the "chain-rule" of the KLD (Thomas & Joy, 2006) into a marginal matching term and a label correctness term:

$$D_{KL}\left(p_{\mathcal{Q}} || p_{\text{cov}}\right) = \underbrace{D_{KL}\left(p_{\mathcal{Q}}(\boldsymbol{c}) || p_{\text{cov}}(\boldsymbol{c})\right)}_{\text{marginal matching}} + \underbrace{\mathbb{E}_{\boldsymbol{c}\sim D_\theta}\left[D_{KL}\left(p_{\mathcal{Q}}(y|\boldsymbol{c}) || p_{\text{cov}}(y|\boldsymbol{c})\right)\right]}_{\text{label correctness}}. \quad (5)$$

Intuitively, the *marginal matching* term requires the synthetic samples' features to be close to the range we are interested in covering. We can compute this term by introducing tractable parametric densities. The *label correctness* term penalizes divergence between the conditional label distributions. Intuitively, it enforces that the samples' labels correspond to their true labels and penalizes uncertainty in the label. We propose to model this label uncertainty using our weak entailment certainty estimates $r$ and provide details on how we model both terms in the following paragraphs.

**Parametrizing densities.** We need to insert a suitable and tractable parametrizations of $p_{\text{cov}}$ and $p_\theta$. We start by modeling their marginals. To model $p_{\text{cov}}$ we chose an efficiently tractable density $p_{\text{cov}}(\boldsymbol{x})$ can be defined via the nearest target feature vector in $\text{claim}(\mathcal{D}_{t,e})$ by[4]

$$p_{\text{cov},\sigma_q}(\boldsymbol{c}) = \frac{1}{Z}\exp\left(-\frac{\|\boldsymbol{c} - \arg\min_{\boldsymbol{c}_i \in \mathcal{D}_t} d(\boldsymbol{c}_i, \boldsymbol{c})\|_2^2}{\sigma_q^2}\right), \quad (6)$$

where $Z > 0$ is a normalization constant. We show that a finite $Z$ always exists in the Appendix B.6. The constant $\sigma_q > 0$ will be treated as a hyperparameter in our framework. Let $\mathcal{Q} \subset \mathcal{X}$ denote a finite set of selected samples. For $p_\theta$, we chose a standard kernel density estimator with kernel width $\sigma_r \geq 0$:

$$p_{\theta(\mathcal{Q}),\sigma_r}(\boldsymbol{c}) = \frac{1}{|\mathcal{Q}|}\sum_{\hat{\boldsymbol{c}}_i \in \mathcal{Q}} \mathcal{N}(\boldsymbol{c}; \hat{\boldsymbol{c}}_i, \sigma_r^2 \boldsymbol{I}). \quad (7)$$

**Modeling label correctness.** We propose to model to label correctness term using the entailment certainty scores, which provides us with an estimate of how well the true and the assigned labels are aligned at a certain point. If a positively labeled sample has very high entailment certainty or a negatively labeled sample has very low entailment certainty, the assigned labels likely match the ground truth and divergence between true conditional label distribution and assumed distribution is expected to be minimal at the sample $\boldsymbol{x}$. We derive a relation between the label correctness term and our entailment certainty score in form of a function $D_{KL}\left(p_\theta(y|\boldsymbol{c}) || p_{\text{cov}}(y|\boldsymbol{c})\right) := \text{LDiv}(r^{(i)}, \hat{y})$, relying on the current entailment certainty $r^{(i)}$ and the assigned hard label $\hat{y}$ in Appendix B.2 that is depicted in Figure 4. The resulting relation fulfills certain natural axioms including that the label correctness term is 0, when we have perfect certainty, i.e., $\text{LDiv}(0, 0) = 0$, $\text{LDiv}(1, 1) = 0$.

## B.2   MODELING AND TRACKING LABEL UNCERTAINTY

In this section we provide a strategy to estimate the label correctness term in Equation (5) which is given by $D_{KL}\left(p_{\mathcal{Q}}(y|\boldsymbol{c}) || p_{\text{cov}}(y|\boldsymbol{c})\right)$ (see Appendix B.1 for why we need to model this term). We

---

[4]we now assume $\boldsymbol{x} \in \mathcal{X}$ to be in a metric space. This can be achieved using an encoder mapping the textual input to real vectors.

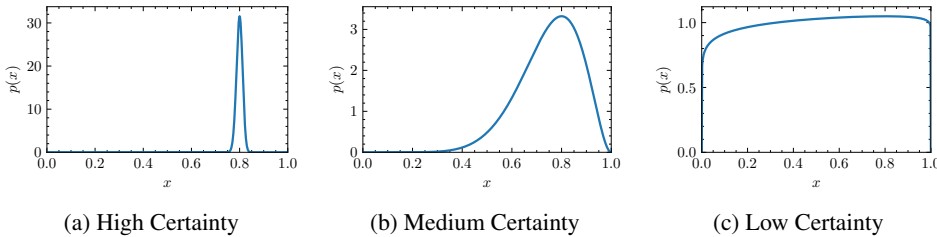

(a) High Certainty      (b) Medium Certainty      (c) Low Certainty

Figure 5: We model the label uncertainty through a hyper distribution over the parameter $\varphi$.

need to model both $p_{\mathcal{Q}}(y|\boldsymbol{c})$ and $p_{\text{cov}}(y|\boldsymbol{c})$ to estimate this term. As they are binary, we choose Bernoulli distributions. Our estimated conditional $p_{\mathcal{Q}}(y = 1|\boldsymbol{c}) = \phi_0$ is modeled through a Bernoulli distribution with parameter $\phi_0$. This conditional distribution is assumed not to change through the augmentation once initialized (because we also do not change the hard labels during augmentations). Reasonable choices for $\phi_0$ involve setting hard probabilities, i.e., $\phi_0 = \hat{y}$ or using the initial label certainty score $\phi_0 = r^{(0)}$ as a softer version.

Unfortunately, we do not directly have access to the true label distribution $p_{cov}(y|\boldsymbol{c})$, but we can follow the following intuition: When we arrive at $r \approx 0.5$ due to many augmentations, this indicates no knowledge about the ground truth label of $\boldsymbol{c}$. However, this does not mean that the ground truth distribution $p_{\text{cov}}(y = 1|\boldsymbol{c}) = 0.5$, for instance the sample can still have a certain label that annotators would agree on. Instead there is uncertainty about this distribution's parameter $p_{\text{cov}}(y = 1|\boldsymbol{c}) = \varphi$. There are different options to model the uncertainty over the true label distribution in this work.

We choose the Beta distribution, which is commonly used as a hyperprior for Benoulli distributions. We chose impose two constraints on the distribution and show that they uniquely define the hyperparameter distribution and have some intuitive properties.

**Proposition 1** *Let $\phi \sim Beta(\alpha, \beta)$ denote a Beta distribution. Let $\phi_0$ be the parameter of the (certain) initial label distribution (usually corresponding to $\hat{y}$) and let $r$ denote the probability of the mutated sample having label $y = 1$ (entailment certainty).*

1. *If $r \in [min(0.5, \phi_0), max(\phi_0, 0.5)]$, there exist unique values for $\alpha', \beta'$ such that $\mathbb{E}[p(y = 1|\boldsymbol{c}; \varphi)] = \mathbb{E}[\phi] = r$ with a mode at $\phi = \phi_0$.*

2. *For $r \rightarrow 0.5$, the distribution $Beta(\alpha', \beta')$ with the values from statement 1 converges to a unit distribution on [0,1] in distribution.*

3. *Using $p_{cov}(y = 1|\boldsymbol{c}) = \varphi$, $\varphi \sim Beta(\alpha', \beta')$, $p_{\mathcal{Q}}(y = 1|\boldsymbol{c}) = \varphi_0$ and the expected KLD over the prior has the closed-form solution*

$$\mathbb{E}_{\varphi}\left[D_{KL}\left[p_{\mathcal{Q}}(y|\boldsymbol{c})||p_{cov}(y|\boldsymbol{c}; \varphi)\right]\right] = \tag{8}$$
$$-H(p_{\mathcal{Q}}(y|\boldsymbol{c})) - p_{\mathcal{Q}}(y = 0|\boldsymbol{c})\psi(\beta') - p_{\mathcal{Q}}(y = 1|\boldsymbol{c})\psi(\alpha') + \psi(\alpha' + \beta'). \tag{9}$$

*In the last statement, $\psi$ denotes the digamma-function and $p_{\mathcal{Q}}(y|\boldsymbol{c}) = Bernoulli(\phi_0)$ is the initially assumed label distribution for synthetic samples.*

See Appendix B.5 for a derivation. The convergence behavior of this scheme to a unit distribution is visualized in Figure 5. Using the above update rule and the uncertainty estimation, we can compute the label correctness term for $r = \mathbb{E}\left[p_{\mathcal{Q}}(y = 1|\boldsymbol{c})\right]$

$$\text{LDiv}(\boldsymbol{r}, \phi_0) = H(\text{Bernoulli}(\phi_0)) - (1 - \phi_0)\psi(\beta'(r)) - \phi_0\psi(\alpha'(r)) + \psi(\alpha'(r) + \beta'(r)) \tag{10}$$

where $\alpha'(r)$ and $\beta'(r)$ are the numerical solutions of Proposition 2. To arrive at the formulation in the main paper, we can plug in $\phi_0 = \hat{y}$, which is the term visualized in Figure 4.

### B.3 OPTIMIZING THE OBJECTIVE

With models for the terms in the objective at hand, we can select a set of most promising samples $\mathcal{Q}$ by solving the discrete sample selection problem

$$\min_{\mathcal{Q} \subset \mathcal{D}_\theta^{(i)}, |\mathcal{Q}|=K} \mathcal{L}_{tot}(p_{\theta(\mathcal{Q}),\sigma_r}, p_{\text{cov},\sigma_q}, f). \tag{11}$$

To make the problem computationally tractable, we are particularly interested in estimators that decompose over the individuals samples present in the set $\mathcal{Q}$. With such a decomposition at hand, each sample is assigned an individual contribution and we simply select $K$ samples with lowest individual to contributions to minimize the objective. We derive the following proposition for decomposing our objective using the parametrized distributions with parameters $\sigma_r, \sigma_q, \mathcal{D}_t, \mathcal{Q}$ for $p_\theta$ and $p_{\text{cov}}$ introduced earlier.

**Proposition 2** *As $\sigma_r^2 \to 0$ while $\sigma_q^2 > 0$ is constant, the objective converges to*

$$\lim_{\sigma_r \to 0} \mathcal{L}_{tot}(p_{\theta(\mathcal{Q}),\sigma_r}, p_{\text{cov},\sigma_q}, f) = C + \sum_{\hat{c}_i, \hat{y}_i, r_i \in \mathcal{Q}} \left[ d(\hat{c}_i, c_{\min,i})^2 + \lambda_d \mathit{LDiv}(r_i, \hat{y}_i) - \lambda_u' U_f(\hat{c}_i, \hat{y}_i) \right] \tag{12}$$

*$C$ is a constant, and $\lambda_d(\sigma_q)$, $\lambda_u'$, hyperparameters,*

$$c_{\min,i} := \operatorname*{arg\,min}_{c' \in claim(\mathcal{D}_{t,e})} d(c', \hat{c}_i), \tag{13}$$

*and LDiv denotes the expected KLD of the conditional distribution (label correctness term) of the objective that can be modeled as a continuous function from the entailment certainty scores $r$.*

We derive this proposition in Appendix B.4. In summary, we show that for small $\sigma_r$ the contribution of a sample to the objective approaches a sum of three parts: The distance to the closest sample from the target claim set for evidence $e$, $claim(\mathcal{D}_{t,e})$ to the claim $\hat{c}$, the label correctness term, and the negative utility. We use the above decomposition in our algorithm, ensuring the objective can be solved highly efficiently in three steps: (1) Computing each samples contribution to $\mathcal{L}_{tot}$, (2) ranking the samples by this contribution, and (3) finally selecting the top-$K$ subset of samples with the lowest contributions.

### B.4 DERIVATION OF PROPOSITION 2

To reduce computational complexity, we use an approximation of our objective that does not feature dependencies between the points in the subset. Then the objective is given by a sum of values for the individual points. To derive this objective, we consider the behavior of the objective for $\sigma_r \to 0$ while keeping a fixed $\sigma_q > 0$. First we note that the normal density $p_{\mathcal{Q},\sigma_r}(c)$ with center $c$ and covariance $\sigma_r^2 I$ converges to a Dirac distribution and for a continuous function $f : \mathbb{R}^d \to \mathbb{R}$ with the filter property:

$$\lim_{\sigma_r \to 0} \int_{\mathbb{R}^N} f(c) p_{\mathcal{Q},\sigma_r}(c) dc = f(c). \tag{14}$$

We now consider the individual terms of the objective.

**Marginal Matching.** Let us start with the marginal matching term of the objective.

$$p_{\mathcal{Q}}(x) = \frac{1}{|\mathcal{Q}|} \sum_{\hat{c}_i \in \mathcal{Q}} \mathcal{N}(c; \hat{c}_i, \sigma_r^2 I) := \sum_{\hat{c}_i \in \mathcal{Q}} \bar{p}_i(c) \tag{15}$$

where $\bar{p}_i(c)$ corresponds to the density of the $i$th mixture component.

$$D_{KL}(p_{\mathcal{Q}}(c) \| p_{\text{cov}}(c)) = \mathbb{E}_{c \sim p_{\mathcal{Q}}}[\log p_{\mathcal{Q}}(c) - \log p_{\text{cov}}(c)] \tag{16}$$

$$= \frac{1}{|\mathcal{Q}|} \sum_{\hat{c}_i \in \mathcal{Q}} \mathbb{E}_{c \sim \bar{p}_i}[\log p_{\mathcal{Q}}(c) - \log p_{\text{cov}}(c)] = \frac{1}{|\mathcal{Q}|} \sum_{\hat{c}_i \in \mathcal{Q}} \mathbb{E}_{c \sim \bar{p}_i}[\log p_{\mathcal{Q}}(c)] - \mathbb{E}_{c \sim \bar{p}_i}[\log p_{\text{cov}}(c)] \tag{17}$$

We need to find good and tractable approximations of both terms. For small $\sigma_r \to 0$, $p_i$ approaches a dirac distribution $\delta_{\boldsymbol{\theta}_i}$ with all mass at the center $\boldsymbol{\theta}_i$. This simplifies the objective to

$$\mathbb{E}_{\boldsymbol{c} \sim \bar{p}_i} [\log p_{\mathcal{Q}}(\boldsymbol{x})] \to \mathbb{E}_{\boldsymbol{c} \sim \bar{p}_i} [\log \bar{p}_i(\boldsymbol{c})] \to \frac{1}{2} d \log \left( (2\pi e) + \sigma_r^2 \right) \to \frac{1}{2} d \log(2\pi e) \quad (18)$$

where $d$ is the dimension of $\boldsymbol{c}$. The second part converges to

$$\mathbb{E}_{\boldsymbol{c} \sim \bar{p}_i} [\log p_{\text{cov}}(\boldsymbol{c})] \to \frac{1}{Z} \left[ \frac{-\|\hat{\boldsymbol{c}}_i - \boldsymbol{c}_{\text{min},i}\|^2}{\sigma_q^2} \right] \quad (19)$$

where

$$\boldsymbol{c}_{\text{min},i} := \underset{\boldsymbol{c}' \in \text{claim}\mathcal{D}_{t,\boldsymbol{e}}}{\arg\min} \ d(\boldsymbol{c}', \hat{\boldsymbol{c}}_i). \quad (20)$$

**Label Correctness Term.** We model the uncertainty propagation as in Equation (10). Approaching $\sigma_r \to 0$, we have

$$\mathbb{E}_{\boldsymbol{c} \sim p_{\mathcal{Q}}} [D_{KL} (p_\theta(y|\boldsymbol{x}) \| p_{\text{cov}}(y|\boldsymbol{c}))] = \mathbb{E}_{\boldsymbol{c} \sim p_\theta} [\text{LDiv}(r, y)] \quad (21)$$

$$\to \frac{1}{|\mathcal{Q}|} \sum_{\hat{\boldsymbol{c}}_i, y_i, r_i \in \mathcal{Q}} [\text{LDiv}(r_i, \hat{y}_i)] \quad (22)$$

**Utility Term** Finally, the same can be done for the utility term:

$$\lambda_u \mathbb{E}_{(\boldsymbol{c}, y) \sim p_{\mathcal{Q}}} [U_f(\boldsymbol{c}, y)] \to \frac{1}{|\mathcal{Q}|} \sum_{\hat{\boldsymbol{c}}_i, \hat{y}_i, r_i \in \mathcal{Q}} \lambda_u \left[ U_f(\hat{\boldsymbol{c}}_i, \hat{y}_i) \right] \quad (23)$$

**Assembling all terms.** In summary, we arrive at

$$\mathcal{L}_{tot} \to \frac{1}{|\mathcal{Q}|} \sum_{\hat{\boldsymbol{c}}_i, \hat{y}_i, r_i \in \mathcal{Q}} \frac{d}{2} \log(2\pi e) + \frac{1}{Z} \left[ \frac{\|\hat{\boldsymbol{c}}_i - \boldsymbol{c}_{\text{min},i}\|^2}{\sigma_q^2} \right] + \text{LDiv}(r_i, \hat{y}_i) - \lambda_u U_f(\hat{\boldsymbol{c}}_i, \hat{y}_i) \quad (24)$$

$$= \frac{d}{2} \log(2\pi e) \sum_{\hat{\boldsymbol{c}}_i, \hat{y}_i, r_i \in \mathcal{Q}} \frac{1}{|\mathcal{Q}| Z \sigma_q^2} \|\hat{\boldsymbol{c}}_i - \boldsymbol{c}_{\text{min},i}\|^2 + \frac{1}{|\mathcal{Q}|} \text{LDiv}(r_i, \hat{y}_i) - \frac{\lambda_u}{|\mathcal{Q}|} U_f(\hat{\boldsymbol{c}}_i, \hat{y}_i) \quad (25)$$

$$\propto C + \sum_{\hat{\boldsymbol{c}}_i, \hat{y}_i, r_i \in \mathcal{Q}} \|\hat{\boldsymbol{c}}_i - \boldsymbol{c}_{\text{min},i}\|^2 + \lambda_d \text{LDiv}(r_i, \hat{y}_i) + \lambda_u' U_f(\hat{\boldsymbol{c}}_i, \hat{y}_i) \quad (26)$$

where in the last step, we multiply all terms be $|\mathcal{Q}| Z \sigma_q^2$ to normalize the first constant to 1. This completes our derivation.

### B.5   DERIVATION OF PROPOSITION 1

**Statement 1:** We know that the mean of the beta distribution $\varphi \sim \text{Beta}(\alpha, \beta)$ is given by

$$\mathbb{E}[\varphi] = \frac{\alpha}{\alpha + \beta} \quad (27)$$

and the mode is given by

$$\text{mode}[\varphi] = \frac{\alpha - 1}{\alpha + \beta - 2} \quad (28)$$

for $\alpha, \beta > 1$ (for $\alpha = \beta = 1$, we obtain the uniform distribution and any value is a mode). Constraining the mode to be $\text{mode}[\varphi] = \phi_0$ yields

$$\alpha = q\phi_0 + 1, \beta = q(1 - \phi_0) + 1, q \in [0, \infty) \quad (29)$$

For the mean, we obtain

$$\mathbb{E}[\varphi] = \frac{q\varphi_0 + 1}{q + 2} \quad (30)$$

Setting $\mathbb{E}[\varphi] = r$ yields

$$(q+2)r = q\varphi_0 + 1 \Leftrightarrow q(r - \varphi_0) = 1 - 2r \Leftrightarrow q = \frac{1 - 2r}{r - \varphi_0} \tag{31}$$

The solution of $q$ is non-negative if $r \neq \varphi_0$ and if $\varphi_0 > r$ in case $r > 0.5$ and if $\varphi_0 < r$ in case $r < 0.5$. Under these conditions, we obtain the unique parameters

$$\alpha = \frac{1 - 2r}{r - \varphi_0}\varphi_0 + 1, \beta = \frac{1 - 2r}{r - \varphi_0}(1 - \varphi_0) + 1 \tag{32}$$

**Statement 2.** We prove this statement using the *Method of Moments* showing that each moment of the distribution converges to the moment of the uniform distribution. If this is the case, the method of moments asserts that the sequence will converge in distribution. Note that both distributions are uniquely determined by their moments because they reside on the interval [0,1]. We see that as $r \to 0.5$ we have that $q \to 0$. We will show that

$$\text{Beta}(q\varphi_0, q(1 - \varphi_0)) \xrightarrow{q \to 0} \text{Unif}[0, 1] \tag{33}$$

We therefore compute the $n$th moments of the Unit distribution for $n \in \mathbb{N}$ with $X \sim \text{Unif}[0, 1]$

$$\mathbb{E}[X^N] = \frac{1}{n + 1} \tag{34}$$

For the Beta distribution with with $\phi \sim \text{Beta}(q\varphi_0, q(1 - \varphi_0))$ we have

$$\mathbb{E}[\varphi^N] = \prod_{k=0}^{n-1} \frac{q\varphi_0 + 1 + k}{q + 2 + k} \tag{35}$$

Taking the limit results in

$$\lim_{q \to 0} \mathbb{E}[\varphi^N] = \lim_{q \to 0} \prod_{k=0}^{n-1} \frac{q\varphi_0 + 1 + k}{q + 2 + k} = \prod_{k=0}^{n-1} \frac{1 + k}{2 + k} = \frac{1}{n + 1} \tag{36}$$

**Statement 3**: We calculate

$$\mathbb{E}_\varphi\left[D_{KL}\left[p_{\mathcal{Q}}(y|\boldsymbol{c})||p_{\text{cov}}(y|\boldsymbol{c}; \varphi)\right]\right] = \tag{37}$$
$$p_{\mathcal{Q}}(y = 0|\boldsymbol{c})(\log p_{\mathcal{Q}}(y = 0|\boldsymbol{c}) - \mathbb{E}_\varphi[\log 1 - \varphi]) \tag{38}$$
$$+ p_{\mathcal{Q}}(y = 1|\boldsymbol{c})(\log p_{\mathcal{Q}}(y = 1|\boldsymbol{c}) - \mathbb{E}_\varphi[\log \varphi]) \tag{39}$$
$$= -H(p_{\mathcal{Q}}(y|\boldsymbol{c})) - p_{\mathcal{Q}}(y = 0|\boldsymbol{c})(\psi(\beta) - \psi(\alpha + \beta)) - p_\theta(y = 1|\boldsymbol{c})(\psi(\alpha) - \psi(\alpha + \beta)) \tag{40}$$
$$= -H(p_{\mathcal{Q}}(y|\boldsymbol{c})) - p_{\mathcal{Q}}(y = 0|\boldsymbol{c})\psi(\beta) - p_\theta(y = 1|\boldsymbol{c})\psi(\alpha) + \psi(\alpha + \beta) \tag{41}$$

We use the identities:

$$\mathbb{E}_{\varphi \sim \text{Beta}(\alpha,\beta)}[\log \varphi] = \psi(\alpha) - \psi(\alpha + \beta) \tag{42}$$
$$\mathbb{E}_{\varphi \sim \text{Beta}(\alpha,\beta)}[\log 1 - \varphi] = \psi(\beta) - \psi(\alpha + \beta) \tag{43}$$

**Definition 1 (Probabilistically Correct Data Augmentation, PCDA)** *A probabilistically correct data augmentation is a (potentially randomized) mapping $M : \mathcal{X} \times \mathcal{Y} \to \mathcal{X} \times [0, 1]$. Applying $(\boldsymbol{x}_{k+1}, r_{miss}) = M(\boldsymbol{x}_k, y)$ generates a modified sample $\boldsymbol{x}_{k+1}$ and additionally returns a probability $r_{miss} \in [0, 1]$ of flipping the assigned label during the augmentation step when keeping the mechanism $g$ and the annotator $\eta$ fixed, i.e., $p(g(\boldsymbol{x}_{k+1}, \eta) \neq g(\boldsymbol{x}_k, \eta)) = r_{miss}^{(k)}$, where the randomness is over the data augmentation output $\boldsymbol{x}_{k+1}$. Setting the initial agreement $p_{agree}^{(0)} = 1.0$, we can perform the following update rule*

$$p_{agree}^{(k+1)} = (1 - r_{miss}^{(k)})p_{agree}^{(k)} + r_{miss}^{(k)}(1 - p_{agree}^{(k)}). \tag{44}$$

### B.6 EXISTENCE OF NORMALIZATION CONSTANT

We consider the density

$$p_{\text{cov}}(\boldsymbol{x}) = \frac{1}{Z} \exp\left(-\frac{\|\boldsymbol{x} - \arg\min_{\boldsymbol{x}_i \in \mathcal{D}_t} d(\boldsymbol{x}_i, \boldsymbol{x})\|_2^2}{\sigma_q^2}\right), \tag{45}$$

where $D_t$ is a finite set. We show that the normalization constant exists by proving that the integral of the non-normalized density over the feature space $\mathcal{X} = \mathbb{R}^d$ is bounded. To do so, we perform the following derivation:

$$\int_{\mathbb{R}^d} \exp\left(-\frac{\|\boldsymbol{x} - \arg\min_{\boldsymbol{x}_i \in \mathcal{D}_t} d(\boldsymbol{x}_t, \boldsymbol{x})\|_2^2}{\sigma_q^2}\right) d\boldsymbol{x} \le \int_{\mathbb{R}^d} \sum_{x_i \in D_t} \exp\left(-\frac{\|\boldsymbol{x} - \boldsymbol{x}_i\|_2^2}{\sigma_q^2}\right) d\boldsymbol{x} \tag{46}$$

$$\le \sum_{x_i \in D_t} \frac{1}{\sqrt{\sigma_q^2 \pi}} \le \frac{|D_t|}{\sqrt{\sigma_q^2 \pi}} \tag{47}$$

The first step uses the insight that the argmin will always be any point in $D_t$ so if we add up the contributions for all possible points, we will arrive at an upper bound. This completes the proof.

## C IMPLEMENTATION DETAILS

### C.1 IMPLEMENTATION DETAILS FOR AUGMENTATION STRATEGIES

In this section we provide additional details regarding the data augmentation strategies that we deploy in this work. We fully commit to open-sourcing our code to reproduce the experiments upon acceptance.

**Partial Rephrasing with LLM.** We use the prompt given in Appendix E.3 to instruct the LLM (Claude3-Haiku) to create different versions of a document where some parts are masked. We decide to mask a random 20% of consecutive words in the document. We let the LLM generate 3 outputs each for 2 different masks, resulting in a total of 6 rephrased versions for each claim. Sampling temperature is set to 1.0.

**Complete Paraphrasing.** We use the T5-based model obtained here[5] as a paraphraser to generate 3 rephrased versions of each claim. To ensure no duplicates are produced, we set parameters `repetition_penalty=10.0`, and `no_repeat_ngram_size=5`.

**Drop Sentences.** For this augmentation, we sentence tokenize the claim using `spacy` with `en_core_web_sm` tokenizer. We postprocess the outputs slightly to better handle statements in quotes. We then randomly drop a sentence from the claim.

### C.2 DATASETS

We apply the following preprocessing to the datasets: We filter out all samples, that have more than 1022 BART tokens (filling out the 1024 context length with an additional SEP and CLS token). The sizes and source links of the resulting datasets are provided in Table 4. We note that this reduces the number of usable SummEdits domains from 10 to 5 (due to some domains only containing overlength evidence documents).

**Splits.** We either use the available train/test splits (RAGTruth) or create splits making sure that summaries / answers derived from the same evidence are either only present in the train or the test split. The validation split is derived from the train split.

**Processing of QA datasets.** The QA datasets require integrating the question and the retrieved documents into a single prompt. The RAG-Truth dataset already provides integrated prompts which we use. For the LFQA-Verification questions and documents are provided seperately. We use the integration template *"You are given the question: " + <QUESTION> Here is some information related to the question: <EVIDENCE DOCUMENTS>*.

---

[5] `https://huggingface.co/humarin/chatgpt_paraphraser_on_T5_base`

| Dataset | Train | Val | Test | Link |
|---|---|---|---|---|
| ragtruth-Summary | 2578 | 125 | 636 | `https://github.com/ParticleMedia/RAGTruth` |
| ragtruth-QA | 3661 | 143 | 875 | `https://github.com/ParticleMedia/RAGTruth` |
| summedits | 2671 | 60 | 733 | `https://huggingface.co/datasets/Salesforce/summedits` |
| lfqa-verification | 171 | 35 | 65 | `https://github.com/timchen0618/LFQA-Verification/` |

Table 4: Dataset sizes.

| Dataset | RAGTruth | | LFQA-Verification | SummEdits |
|---|---|---|---|---|
| Parameter / RAG-Task | Summary | QA | QA | Summary |
| # Samples per evidence | 8 | 8 | 4 | 32 |
| # Synth. Dataset size (Org. Size) | 3544 (2578) | 5032 (3662) | 336 (171) | 3552 (2671) |
| # Augmentation Iterations | 2 | 2 | 1 | 1 |

Table 5: Fixed hyperparameters dependent on dataset. Note that we set only "Samples per evidence" which determines synthetic dataset size together with the number of evidences.

### C.3 AUTO-GDA DETAILS AND HYPERPARAMETERS

We implement the algorithm outlined in Algorithm 1. We emphasize that we fix the teacher model to assign the initial scores. Here we can compute estimates of the model performance using the validation set of evidence-claim pairs to which we have access, allowing us to chose the best performing one as teacher. However, we do not know the performance of the models on claim-claim pairs, so we treat the teacher model used in the augmentation step as a hyperparameter, that will be optimized.

**Fixed Hyperparameters.** We additinally keep the following hyperparameters fixed across datasets:

- Finetuning: 1 Epoch, learning rate $10^{-5}$ for DeBERTA, BART, $2 \times 10^{-4}$ for FLAN-T5, batch size 2

- To compute the distance function $d$ we use embeddings from a `sentence-t5-base` model[6].

- Number of offspring per sample ($l$ in pseudocode): $l = 12$, with 6 child samples from LLM Partial rephrasing, 3 each from Drop Sentence and Complete Paraphrasing

We set the number of evidences used per claim which determines size of the synthetic dataset according to the different datasets as given in Table 5. Setting our chosen values results in the synthetic dataset being between 1.3-times and 2 times as large as the original dataset based on the oberservations in Appendix D.6, suggesting that this is the optimal range.

**Optimized Hyperparameters.** As outlined in the main text, we apply *optuna* for 50 configuration trial as an hyperparameter optimizer to find the remaining hyperparameters. We set the ranges $\lambda_u \in [0.1, 100], \lambda_u \in [0.01, 5000]$ and let the augmentation teacher model be selected from {vectara, alignscore, deberta}. We don't allow LLMs as teacher models for augmentations because it would be too expensive as a lot of augmented samples are created in the course of the algorithm. The final hyperparameters found through optimization are given in Table 6.

**Base models.** As a basis for finetuning we use huggingface checkpoints for DeBERTaV2[7], BART-large[8] and FLAN-T5[9].

**Hardware.** Our experiments (including runtime) were run on a system with 16-core Intel(R) Xeon(R) CPU E5-2686 processors (2.30GHz) and a single Nvidia Tesla V100 GPU with 32GB of RAM.

---

[6]`https://huggingface.co/sentence-transformers/sentence-t5-base`
[7]`https://huggingface.co/tasksource/deberta-base-long-nli`
[8]`https://huggingface.co/facebook/bart-large-mnli`
[9]`https://huggingface.co/sjrhuschlee/flan-t5-base-mnli`

| Dataset/Task | Initial Teacher Model | Augment. Teacher Model | $\lambda_d$ | $\lambda_u$ |
|---|---|---|---|---|
| | Main Results, Table 1 best NLI model as initial teacher) | | | |
| **RAGTruth-QA** | **vectara** | debertav2 | 32.67 | 20.57 |
| **RAGTruth-Summ** | **vectara** | debertav2 | 198.85 | 19.51 |
| **LFQA-QA** | **alignscore** | vectara | 25.27 | 6.83 |
| **Summedits-Summ** | **alignscore** | bart-large | 0.02 | 92.11 |
| | GPT-4o teacher results, Table 9 (GPT-4o as initial teacher) | | | |
| **RAGTruth-Summ** | **gpt-4o** | vectara | 0.06 | 7.58 |
| **RAGTruth-QA** | **gpt-4o** | bart-large | 47.26 | 0.15 |
| **LFQA-QA** | **gpt-4o** | debertav2 | 3940.60 | 7.28 |
| **Summedits-Summ** | **gpt-4o** | debertav2 | 0.01 | 29.42 |
| | Self-supervised results, Table 9 (DeBERTa as initial, augmentation teacher) | | | |
| **RAGTruth-Summ** | **debertav2** | **debertav2** | 296.13 | 1.03 |
| **RAGTruth-QA** | **debertav2** | **debertav2** | 4591.98 | 0.24 |
| **LFQA-QA** | **debertav2** | **debertav2** | 890.45 | 17.29 |
| **Summedits-Summ** | **debertav2** | **debertav2** | 871.77 | 19.23 |

Table 6: Tuned hyperparameters. **Bold** parameters were fixed for the runs, while the remainder was tuned using the hyperparameter optimizer.

## C.4   ALGORITHM

---

**Algorithm 1** Automatic Generative Domain Adaptation (Auto-GDA)

---

**Require:**  Set of target features $\mathcal{D}_t$, no. best neighbors to select $K$, no. of augmentations per sample $l$, Generator $G$, augmentation modification function $M$, teacher model $T$, base model $f$

1:  $D = \{\}$
2:  **for** Each unique evidence $\boldsymbol{e}$ in $D_t$ **do**:
3:      $\hat{y}_k \leftarrow \text{Bernoulli}(0.5), \forall k = 1...K$                                  $\triangleright$ Sample $K$ labels
4:      $\hat{\boldsymbol{c}}_k \leftarrow G(\boldsymbol{e}, \text{claim}(D_{t,\boldsymbol{e}}), \hat{y}_k), \forall k = 1...K$          $\triangleright$ Sample initial claims using generator $\mathcal{G}$
5:      $r_k^{(0)} \leftarrow T(\boldsymbol{c}_k, \hat{y}_k), \forall k = 1...K$                          $\triangleright$ get their label probabilities $\boldsymbol{r}^{(0)}$.
6:      $D_{\boldsymbol{e}}^{(0)} = \left\{ \hat{\boldsymbol{c}}_k, \hat{y}_k, r_k^{(0)} \right\}_{k=1}^{K}$
7:      $i \leftarrow 0$
8:      **while** $\mathcal{L}_{tot}(D_{\boldsymbol{e}}^{(i)})$ has not converged **do**
9:          $i \leftarrow i + 1$
10:         $\bar{D}_{\boldsymbol{e}}^{(i)} \leftarrow D_{\boldsymbol{e}}^{(i-1)}$
11:         **for** $(\hat{\boldsymbol{c}}_k, \hat{y}_k, r_k^{(i-1)}) \in (D_{\boldsymbol{e}}^{(i-1)})$ **do**
12:             **for** $l$ times **do**
13:                 $\hat{\boldsymbol{c}}' = M(\hat{\boldsymbol{c}}_k)$                      $\triangleright$ Augment sample through mutation function
14:                 $r^{(i)} \leftarrow r_k^{(i-1)}(\boldsymbol{e}, \hat{\boldsymbol{c}}_k) \cdot T(\hat{\boldsymbol{c}}_k, \hat{\boldsymbol{c}}') + (1 - r_k^{(i-1)}(\boldsymbol{e}, \hat{\boldsymbol{c}}_k)) \cdot (1 - T(\hat{\boldsymbol{c}}_k, \hat{\boldsymbol{c}}'))$
15:                 $\bar{D}_{\boldsymbol{e}}^{(i)} \leftarrow \bar{D}_{\boldsymbol{e}}^{(i)} \cup \{(\hat{\boldsymbol{c}}', r^{(i)}, \hat{y}_k)\}$                      $\triangleright$ update $r$ and append sample
16:             **end for**
17:         **end for**
18:         $D_{\boldsymbol{e}}^{(i)} \leftarrow \arg\min_{\mathcal{Q} \subset \bar{D}_{\boldsymbol{e}}^{(i)}, |\mathcal{Q}| = K} \mathcal{L}_{\text{tot}}(\mathcal{Q})$                      $\triangleright$ Select best sample subset
19:     **end while**
20:     $D \leftarrow D \cup D_{\boldsymbol{e}}^{(i)}$
21: **end for**
22: $f' = \text{fine-tune}(f, D)$                      $\triangleright$ Fine-tune model $f$ on synthetic dataset $D$
23: **return** $f'$

---

We provide pseudocode for our algorithm in Algorithm 1.

### C.5 BASELINE NLI MODELS

For the complex NLI model baselines, we use Vectara HHEM-2.1[10]. The model cannot be easily fine-tuned because it uses custom code. Additionally we use Alignscore-base with the checkpoint found in this repository[11] with the recommended "split" (pre- and postprocessing) `nli_sp` option. We neglect the larger version as its runtime was comparable to LLMs at a usually lower performance, making the smaller model a better trade-off. Finally we use the best-performing Minicheck `flan-t5-large` model by Tang et al. (2024) from the official huggingface page[12].

### C.6 ROBUST PRE-TRAINING AND FINE-TUNING FOR UNSUPERVISED DOMAIN ADAPTATION

To address domain shift in NLI, we experimented with multiple classical unsupervised domain adaptation (UDA) techniques, which aim to improve the model's generalization for out-of-domain data by adding robustness during training and by using unlabeled target-domain data. Specifically, we implemented Domain-Adaptive Pretraining (DAPT), Virtual Adversarial Training (VAT), Deep CORrelation ALignment (CORAL), and Domain Adversarial Neural Networks (DANN) combined with conditional entropy minimization.

**Notation** We refer to source domain data $\{(x_i, y_i)\}_{i=1:n} \in (X_S, Y_S)^n$, where $x_i$ denotes the input text and $y_i \in \{0, 1\}$ the corresponding entailment label, and target domain data $\{(x_i)\}_{i=1:m} \in (X_T)^m$. The labels $y \in \{0, 1\}$ correspond to whether a claim is entailed or hallucinated (contradictory or neutral). Our goal is to train a feature extractor $f_\theta$, parameterized by $\theta$, that performs well on the target domain.

**Domain-Adaptive Pretraining (DAPT)** Our first approach, Domain-Adaptive Pretraining (DAPT) (Gururangan et al., 2020), performs Masked Language Modeling (MLM) on (unlabeled) data from the target domain before finetuning on the labeled source-domain data for NLI. This way, it learns the representations of both source and target domain, before finetuning on the source domain to relearn classification for the NLI task.

**Virtual Adversarial Training (VAT)** Our second approach is Virtual Adversarial Training (Miyato et al., 2016), which increases model robustness by introducing adversarial perturbations to the input data during training. Specifically, we add an adversarial regularization term to the classification objective, which becomes:

$$\min_\theta \mathbb{E}_{(x,y) \sim D_S}[\ell(f_\theta(x), y)] + \lambda \cdot \mathbb{E}_{x \sim D_S} \left[ \max_{\delta \in S} \ell_{\text{KL}}(f_\theta(x + \delta), f_\theta(x)) \right], \tag{48}$$

where $\ell$ is the classification loss (e.g., cross-entropy) on the source domain, $\ell_{\text{KL}}$ is the KL divergence between the output distributions, $\delta$ is a small perturbation constrained within a a ball $S$, and $\lambda$ is the regularization weight. According to (Jiang et al., 2020), VAT induces Lipschitz-continuity, which means that small changes in the input do not cause disproportionately large changes in the output, improving robustness and generalization for out-of-domain data.

**Deep CORAL** The objective of CORrelation ALignment (CORAL) (Sun et al., 2016) is to align the second-order statistics (covariances) of the source and target embedding distributions by minimnizing the Frobenius norm between their covariance matrices. Specifically, denoting $\mathbf{C}_S$ and $\mathbf{C}_T$ the covariance matrices of the embeddings of the source and target samples as extracted from the last encoding layer, respectively, and as $d$ the dimension of the features, the regularization loss is:

$$L_{\text{CORAL}}(\theta) = \frac{1}{4d^2} \|\mathbf{C}_S - \mathbf{C}_T\|_F^2, \tag{49}$$

**Domain Discriminator and Conditional Entropy Minimization** Finally, we also implement a domain discriminator inspired by domain adversarial training (Ganin et al., 2016) and other related works in NLP. The discriminator $D$ is trained to classify source and target domain features correctly,

---

[10]`https://huggingface.co/vectara/hallucination_evaluation_model`
[11]`https://github.com/yuh-zha/AlignScore`
[12]`https://huggingface.co/lytang/MiniCheck-Flan-T5-Large`

whereas the feature extractor (classifier) $f_\theta$ is trying to minimize the discriminator's accuracy, which should amplify the learning of domain-invariant features from the classifier. The domain adversarial loss is:

$$L_d(\theta) = \mathbb{E}_{x \sim D_S}[\ln D(f_\theta(x))] + \mathbb{E}_{x \sim D_T}[\ln(1 - D(f_\theta(x)))]. \tag{50}$$

We also experiment with adding a conditional entropy loss to ensure the model makes confident predictions on the target domain and improve the placement of the initial boundaries, as outlined in Shu et al. (2018) and Reed et al. (2014):

$$L_c(\theta) = -\mathbb{E}_{x \sim D_T}\left[f_\theta(x)^\top \ln f_\theta(x)\right], \tag{51}$$

**Implementation Details** For our robust optimization experiments we used the DeBERTaV2 based NLI model, and limited maximum tokenization length at 1,024 tokens across all benchmarks. For DAPT we extracted the DeBERTaV2 backbone and trained on the target domain for 1 full epoch, using 10% masking probability. At the fine-tuning stage of both methods we ran 1 full epoch on the MNLI dataset. At the masked pre-training and fine-tuning stages of the experiments, we used the AdamW optimizer with learning rate $10^{-5}$ and weight decay $10^{-3}$, enabling 100 warm-up steps over the supervised fine-tuning. For SiFT and CORAL we set the coefficient of the respective regularization terms to 0.5, after running hyperparameter optimization with a coarse grid search. Batch size used for covariance estimation in CORAL was set at 64. Each experiment was repeated over 3 random trials.

# D ADDITIONAL RESULTS

## D.1 ADDITIONAL METRICS

We chose the Area under Curve for Receiver-Operator-Characteristic (AUC-ROC) as our main metric, as it is less dependent on threshold calibration and also works for imbalanced datasets. We report our results in other metrics such as balanced accuracy without threshold calibration (using 0.5. as a threshold as suggested in Tang et al. (2024)) in Table 7 and F1-Scores in Table 8. The results highlight not only that our main results are valid across different metrics – in uncalibrated balanced accuracy, our models trained with Auto-GDA data even outperform LLMs by an average 3.4 accuracy percent points.

## D.2 ADDITIONAL QUALITATIVE RESULTS

**Estimating the mislabeling probability.** An integral part of our algorithm in the estimation of the agreement probability in Equation (44). To investigate the effect of implementing this choice, we run an ablation study to better understand how the quality of the agreement probabilities affects the score. We provide results average over 3 runs in Figure 7a. The results indicate that the choice of the model used to estimate $r$ the label certainty score has substantial effect on the quality of the results. While the utility (in terms of ROC scores) drops when noise is added, it increases again when high levels of noise are applied. We attribute this behavior to the algorithm neglecting the mutated samples almost entirely when the noise level is too high and mainly selecting the few shot generate samples. As shown in Table 2 these have fair utility already.

**Using one vs. several augmentation routines.** A key design goal of our algorithm was the ability to automatically select the most promising augmented samples. We investigate the effect of using only single or several augmentations in Figure 6a. The results highlight that the Partial Rephrasing augmentation with LLMs as well as the sentence deletion augmentation seems to be most successful. The Complete Paraphrasing augmentation leads to substantially lower data quality on its own. However, the best utility is achieved when all three augmentations are combined. We study the origin of the samples eventually selected by our algorithm and find that the usefulness of the augmentations on their own is reflected by the share samples selected from each of the augmentations as depicted in Figure 6b. Together, this highlights that Auto-GDA succeeds in selecting the most promising samples generated from the augmentations automatically.

| Dataset | RAGTruth | | LFQA-Verif. | SummEdits | Avg. |
|---|---|---|---|---|---|
| RAG-Task | Summary | QA | QA | Summary | |
| **base models** | | | | | |
| FLAN-T5 | 0.666 | 0.636 | 0.618 | 0.646 | 0.641 |
| DeVERTaV2 | 0.727 | 0.505 | 0.588 | 0.810 | 0.658 |
| BART-large | 0.604 | 0.633 | 0.782 | 0.625 | 0.661 |
| **robustness** | | | | | |
| DAPT$_{DeBERTaV2}$ | 0.677 $\pm$ 0.004 | 0.654 $\pm$ 0.003 | 0.748 $\pm$ 0.076 | 0.792 $\pm$ 0.005 | 0.718 $\pm$ 0.022 |
| SiFT$_{DeBERTaV2}$ | 0.716 $\pm$ 0.009 | 0.562 $\pm$ 0.006 | 0.810 $\pm$ 0.035 | 0.806 $\pm$ 0.015 | 0.724 $\pm$ 0.016 |
| CORAL$_{DeBERTaV2}$ | 0.657 $\pm$ 0.001 | 0.637 $\pm$ 0.002 | 0.815 $\pm$ 0.001 | 0.792 $\pm$ 0.001 | 0.725 $\pm$ 0.001 |
| **complex** | | | | | |
| MiniCheck-T5 | 0.675 | 0.600 | 0.564 | 0.679 | 0.630 |
| AlignScore | 0.572 | 0.650 | 0.594 | 0.770 | 0.646 |
| Vectara-2.1 | 0.662 | 0.744 | 0.618 | 0.581 | 0.651 |
| **Auto-GDA** | | | | | |
| Flan-T5 (Auto-GDA) | 0.650 $\pm$ 0.005 | 0.703 $\pm$ 0.019 | 0.669 $\pm$ 0.016 | 0.761 $\pm$ 0.020 | 0.696 |
| BART (Auto-GDA) | 0.710 $\pm$ 0.028 | 0.794 $\pm$ 0.011 | 0.772 $\pm$ 0.023 | 0.798 $\pm$ 0.014 | 0.769 |
| DeBERTaV2 (Auto-GDA) | 0.737 $\pm$ 0.009 | 0.784 $\pm$ 0.011 | 0.776 $\pm$ 0.012 | 0.817 $\pm$ 0.009 | 0.778 |
| **LLMs** | | | | | |
| GPT-4o | 0.691 | 0.764 | 0.688 | 0.835 | 0.744 |
| GPT-4o-mini | 0.666 | 0.684 | 0.625 | 0.832 | 0.702 |
| GPT-3.5 | 0.593 | 0.586 | 0.611 | 0.723 | 0.629 |

Table 7: Performance comparison to baselines (uncalibrated balanced accuracy). In this metrics, our models even outperform LLM baselines.

| Dataset | RAGTruth | | LFQA-Verif. | SummEdits | Avg. |
|---|---|---|---|---|---|
| RAG-Task | Summary | QA | QA | Summary | |
| **base models** | | | | | |
| FLAN-T5 | 0.890 | 0.900 | 0.705 | 0.550 | 0.761 |
| DeVERTaV2 | 0.897 | 0.899 | 0.705 | 0.748 | 0.812 |
| BART-large | 0.893 | 0.900 | 0.846 | 0.641 | 0.820 |
| **robustness** | | | | | |
| DAPT$_{DeBERTaV2}$ | 0.655 $\pm$ 0.001 | 0.489 $\pm$ 0.016 | 0.748 $\pm$ 0.076 | 0.762 $\pm$ 0.004 | 0.664 $\pm$ 0.027 |
| SiFT$_{DeBERTaV2}$ | 0.704 $\pm$ 0.019 | 0.362 $\pm$ 0.055 | 0.810 $\pm$ 0.036 | 0.762 $\pm$ 0.007 | 0.660 $\pm$ 0.029 |
| CORAL$_{DeBERTaV2}$ | 0.556 $\pm$ 0.001 | 0.487 $\pm$ 0.022 | 0.815 $\pm$ 0.001 | 0.752 $\pm$ 0.001 | 0.653 $\pm$ 0.006 |
| **complex** | | | | | |
| MiniCheck-T5 | 0.897 | 0.901 | 0.743 | 0.682 | 0.806 |
| AlignScore | 0.888 | 0.903 | 0.847 | 0.744 | 0.846 |
| Vectara-2.1 | 0.910 | 0.921 | 0.702 | 0.498 | 0.758 |
| **Auto-GDA** | | | | | |
| Flan-T5 (Auto-GDA) | 0.901 $\pm$ 0.001 | 0.905 $\pm$ 0.002 | 0.699 $\pm$ 0.006 | 0.693 $\pm$ 0.015 | 0.800 |
| BART (Auto-GDA) | 0.910 $\pm$ 0.003 | 0.923 $\pm$ 0.005 | 0.857 $\pm$ 0.017 | 0.725 $\pm$ 0.010 | 0.854 |
| DeBERTaV2 (Auto-GDA) | 0.912 $\pm$ 0.002 | 0.930 $\pm$ 0.004 | 0.854 $\pm$ 0.014 | 0.750 $\pm$ 0.007 | 0.861 |
| **LLMs** | | | | | |
| GPT-4o | 0.929 | 0.914 | 0.848 | 0.782 | 0.869 |
| GPT-4o-mini | 0.918 | 0.909 | 0.767 | 0.764 | 0.840 |
| GPT-3.5 | 0.887 | 0.899 | 0.746 | 0.687 | 0.805 |

Table 8: Performance comparison to baselines (Binary F1-Scores).

| Dataset | RAGTruth-QA | | | |
|---|---|---|---|---|
| Model | DeBERTa | BART-large | Flan-T5 | mean |
| No Augmentation | 0.836 | 0.845 | 0.772 | 0.818 |
| LLMPartialRephrase | 0.869 | 0.890 | 0.767 | 0.842 |
| Complete Paraphrasing | 0.845 | 0.863 | 0.711 | 0.806 |
| DropSentence | 0.868 | 0.872 | 0.758 | 0.833 |
| All | 0.872 | 0.886 | 0.806 | 0.855 |

(a) Testing the effect of using one vs. several augmentations. On average the best results are obtained when combining several augmentations.

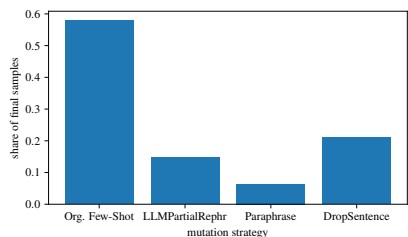

(b) Composition of the final dataset by origin of the selected samples (augmentation routines or few-shot prompting without augmentation). The selection corresponds well to the usefulness of the augmentations on their own.

Figure 6: Qualitative Results on sample selection: Our framework succeeds to automatically select the best samples from different augmentation strategies outperforming single augmentation strategies.

| Dataset | RAGTruth | | LFQA-Verif. | SummEdits | Average |
|---|---|---|---|---|---|
| RAG-Task | Summary | QA | QA | Summary | |
| Teacher Model Used | Vectara-2.1 | Vectara-2.1 | AlignScore | AlignScore | |
| Teacher Performance | 0.805 | 0.854 | 0.904 | 0.894 | 0.864 |
| DeBERTaV2 (Auto-GDA) | 0.837 ±0.007 | 0.867 ±0.007 | 0.925 ±0.009 | 0.883 ±0.005 | 0.878 |
| Teacher Model Used | GPT-4o | GPT-4o | GPT-4o | GPT-4o | |
| GPT-4o Performance | 0.892 | 0.865 | 0.896 | 0.880 | 0.883 |
| DeBERTaV2 (Auto-GDA) | 0.808 ±0.017 | 0.855 ±0.003 | 0.910 ±0.019 | 0.887 ±0.004 | 0.865 |
| Techer Model Used | DeBERTaV2 | DeBERTaV2 | DeBERTaV2 | DeBERTaV2 | |
| DeBERTaV2 Performance | 0.782 | 0.530 | 0.645 | 0.876 | 0.708 |
| DeBERTaV2 (Auto-GDA) | 0.830 ±0.009 | 0.807 ±0.010 | 0.923 ±0.012 | 0.890 ±0.007 | 0.863 |

Table 9: Using different teacher models in Auto-GDA to fine-tune DeBERTaV2. In the upper part we add best results from Table 1 for comparison. In the center part, we highlight that using GPT-4o as a teacher model to assign intial probabilities does not yield substantial improvement. However the lower part shows that it is possible to do self-improvement using only DeBERTa as teacher model for both initial scores and augmentation scoring.

### D.3 DIFFERENT TEACHER MODELS

We investigate the application of different teacher models in Table 9. Our results indicate the learning from GPT models works in general, but does not results in better performance that using the best non-LLM teacher. We additionall study self-improvement, using DeBERTa as both a teacher model for initial scoring and augmentation scoring. This shows that improvements thought self-supervision are possible.

### D.4 ROBUSTNESS TO INITIAL DATA QUALITY

The initial data plays an essential role in Auto-GDA, however we designed our algorithm to be as fault-tolerant as possible and to be able to cope with some low-quality samples in the initial population. For instance, we do not solely rely on generative models, but use discriminative teacher models in the generation loop as well. The selection objective is specifically designed to filter out low-quality samples. Mislabeled samples in the fine-tuning dataset can severely affect performance of a fine-tuned model (Wang et al., 2023b). To test the robustness of Auto-GDA, we manually flip 50 % of the labels in the initial data to study its effect and trace these samples through the generation process. We use the LFQA-Verification dataset and perform 5 runs with different initial generation

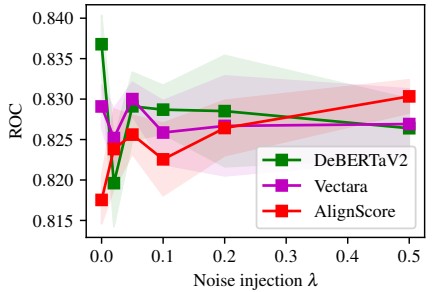 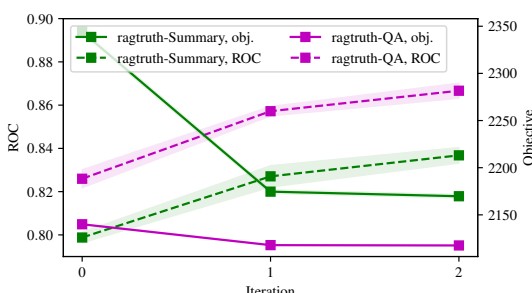

(a) Effect of computing $r$ using different models and noise levels inserted in the scores. While noisy, the results point to different trends for the teacher models: For the best teacher model, noise hurts performance, whereas for the other ones, it does not hurt or might even boost performance as more few-shot samples are selected.

(b) Evolution of ROC scores and our objective over three iterations of our algorithm. We see that ROC increases as our objective decreases. We stop our algorithm after two iterations, when the objective does not improve anymore.

Figure 7: a) Reliable estimation of label certainty $r$ is essential for selection of high quality data. b) The resulting synthetic data often contains original few-shot generated samples as well as a fair mix of mutated samples generated from them. They are automatically selected by our algorithm.

| Dataset | RAGTruth | | LFQA-Verification | SummEdits |
| RAG-Task | Summary | QA | QA | Summary |
|---|---|---|---|---|
| AlignScore | 0.737 | 0.836 | 0.870 | 0.874 |
| Vectara-2.1 | 0.814 | 0.879 | 0.879 | 0.805 |
| GPT-4o | 0.828 | 0.866 | 0.876 | 0.878 |
| Our Data | $0.837 \pm 0.007$ | $0.867 \pm 0.007$ | $0.925 \pm 0.009$ | $0.883 \pm 0.005$ |

Table 10: Comparing our approach to the naive baseline of pseudo-labeling the training data and fine-tuning the DeBERTa V2 model on the pseudo-labeled data.

seeds (using the same hyperparameters as in Table 1 otherwise). First, while 50% of the data have flipped labels initially, in the data selected after one iteration of Auto-GDA, only **10.0% +- 1.1%** are the initial data or augmentations of the data with the flipped labels. After fine-tuning on the generated data, the ROC-AUC score drops slightly by 1.1 points as shown in table below.

| | Original Performance | with 50% flipped initial labels |
|---|---|---|
| ROC-AUC of fine-tuned DeBERTaV2 | $0.925 \pm 0.009$ | $0.914 \pm 0.007$ |

Despite the substantial amount of mislabeled data (50 %) the drop in performance remains small, highlighting that Auto-GDA is quite robust to data quality issues due to its fault-tolerant design.

### D.5 SIMPLE BASELINES

We compare our results to model trained on pseudo-labels in for the original datasets in Table 10. The results inicate that this is a surprisingly strong baseline, which is however surpassed by Auto-GDA in 3 out of 4 cases.

Results when the models are fine-tuned on the validations set directly are shown in Table 12.

### D.6 EFFECT OF DATASET SIZE

When choosing the dataset size we used the data size slightly larger that that of the original dataset as an orientation. We experiment with different dataset sizes and learning rates as shown in Figure 8

| Dataset RAG-Task | RAGTruth Summary | QA | LFQA-Verif. QA | SummEdits Summary | Average |
|---|---|---|---|---|---|
| FLAN-T5 | 0.734 | 0.708 | 0.655 | 0.700 | 0.699 |
| Flan-T5 (Auto-GDA) | 0.756 ± 0.004 | 0.783 ± 0.013 | 0.687 ± 0.002 | 0.824 ± 0.010 | 0.762 (+0.063) |
| BART-large | 0.696 | 0.670 | 0.821 | 0.769 | 0.739 |
| BART (Auto-GDA) | 0.813 ± 0.009 | 0.867 ± 0.011 | 0.867 ± 0.026 | 0.860 ± 0.010 | 0.852 (+0.113) |
| DeBERTaV2 | 0.782 | 0.530 | 0.645 | 0.876 | 0.708 |
| DeBERTaV2 (Auto-GDA) | 0.837 ± 0.007 | 0.867 ± 0.007 | 0.925 ± 0.009 | 0.883 ± 0.005 | 0.878 (+0.170) |

Table 11: Direct comparision of improvements

| Dataset RAG-Task | RAGTruth Summary | QA | LFQA-Verif. QA | SummEdits Summary | Mean |
|---|---|---|---|---|---|
| Non-Fintuned | 0.782 | 0.530 | 0.645 | 0.876 | 0.708 |
| Few-Shot Data | 0.799 | 0.826 | 0.934 | 0.872 | 0.858 |
| FT on validation | 0.784 | 0.750 | 0.899 | 0.890 | 0.833 |
| DeBERTaV2 (Auto-GDA, best teacher) | 0.837 | 0.867 | 0.925 | 0.890 | 0.878 |

Table 12: Fine-tuning on validation set does increase performance but does not reach Auto-GDAs performance on average.

When keeping training fixed to one epoch, we find that with higher learning rates, smaller dataset sizes lead to higher performance, and with lower learning rate, more data is required which seems natural. Globally, we observe that a learning rate of $10^{-5}$ is near optimal, but the performance is rather insensitive This is based on a prior observation that significant oversampling of the dataset size had seemingly little effect.

We also study the effect of generating more LLM samples on the performance of the baseline of solely fine-tuning on LLM-generated data. We generate more LLM data using our few-shot prompting strategy and fine-tune the DeBERTa model on LLM-generated datasets in the range of 0.5x - 8x the size of the datasets used in the main paper. We obtain the results shown in Table 13 (ROC-Scores of the DeBERTaV2 model fine-tuned on the data). Perhaps surprisingly, increasing the number of generated samples does not increase the performance of fine-tuned models. We attribute this to the observation that LLMs (even when setting a higher temperature) will generate highly similar samples after a certain dataset size is reached, not further improving performance of a fine-tuned model. Performance even decreases again, potentially due to overfitting on the synthetic data. We therefore conclude that even with more LLM data, the performance does not reach the level of Auto-GDA.

## D.7 ABLATING TERMS IN THE SELECTION CRITERION

To better understand the effect of the different terms in our objective, we perform an additional ablation study only using a selection objective with one term at a time. We provide the results

| dataset size | 0.5x | 1x | 2x | 4x | 8x |
|---|---|---|---|---|---|
| LFQA | 0.916 ± 0.007 | 0.924 ± 0.006 | 0.903 ± 0.013 | 0.896 ± 0.007 | 0.868 ± 0.003 |
| SummEdits | 0.880 ± 0.003 | 0.876 ± 0.002 | 0.876 ± 0.005 | 0.868 ± 0.002 | 0.855 ± 0.005 |
| RAGTruth-Summary | 0.821 ± 0.003 | 0.808 ± 0.005 | 0.791 ± 0.007 | 0.797 ± 0.001 | 0.783 ± 0.004 |
| RAGTruth-QA | 0.808 ± 0.007 | 0.828 ± 0.006 | 0.830 ± 0.006 | 0.834 ± 0.004 | 0.821 ± 0.005 |
| average | 0.860 | 0.856 | 0.849 | 0.844 | 0.841 |

Table 13: Results when fine-tuning on more LLM generated data (ROC-AUC scores for DeBERTaV2) show that performance of this baseline does not improve with more data.

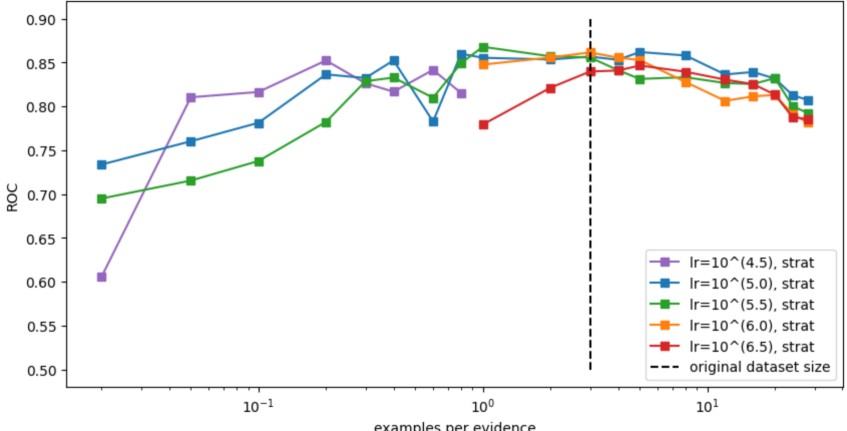

Figure 8: Testing different synthetic dataset sizes and learning rates for fine-tuning. While the original dataset size is at about 3 claim examples per evidence (dashed line) we test a wide range of dataset sizes ranging from 1/50 examples per evidence (100x smaller than original) to larger datasets with 30 claims per evidence, (10x larger than original). We observe a relatively stable optimum from about a 1/3 of original size to twice the original size using learning rates around $10^{-5}$.

| dataset | all terms | only distance | only label-cert. | only utility |
|---|---|---|---|---|
| LFQA | $0.925 \pm 0.004$ | $0.916 \pm 0.004$ | $0.919 \pm 0.003$ | $0.917 \pm 0.005$ |
| Summedits | $0.883 \pm 0.002$ | $0.877 \pm 0.004$ | $0.866 \pm 0.002$ | $0.881 \pm 0.002$ |
| RAGTruth-Summary (1 run) | 0.861 | 0.805 | 0.860 | 0.587 |
| RAGTruth-QA (1 run) | 0.827 | 0.756 | 0.814 | 0.823 |

Table 14: Ablation on terms in the selection criterion (ROC-AUC scores for DeBERTaV2). Using all three terms results in better performance than only using one of the terms each.

in Table 14. Despite some difference being small, we see that all terms contribute to the final performance.

## D.8 EFFECT OF SOURCE DATA ON UDA EFFECTIVENSS

Table 15 compares the performance (ROC-AUC scores) of various Unsupervised Domain Adaptation (UDA) methods and synthetic data approaches across different source datasets. All results are evaluated on the RAGTRUTH target dataset, using a DeBERTaV3 model trained on the PAWS, VitaminC, and Fever data. The table illustrates the effectiveness of different UDA methods and synthetic data approaches when applied to various source datasets (MNLI, Summedits, and Ragtruth-synth). More specifically, we see that except for the synthetically generated version of RAGTruth, the choice of the source domain data does not seem to alter results significantly. We also see that vanilla fine-tuning on the synthetic RAGTruth data outperforms all other variations, indicating that synthetic data is more appropraite for NLI than traditional UDA methods. This is perhaps due to the fact that very small changes in the generated claim can flip the label from entailed to non-entailed and vice-versa.

| Method | Fever+PAWS+VitaminC | Summedits | Synthetic RAGTRUTH |
|---|---|---|---|
| No fine-tuning | 0.735 | 0.735 | 0.735 |
| Vanilla fine-tuning | 0.735 | *0.737* | **0.844** |
| CORAL | 0.682 | 0.683 | 0.728 |
| SMART | *0.743* | 0.721 | 0.833 |
| MLM | 0.680 | 0.577 | 0.731 |
| Domain Discriminator | 0.603 | 0.712 | 0.746 |

Table 15: ROC-AUC scores for different UDA methods and synthetic data approaches.

## D.9   EFFECT OF REGULARIZATION CONSTANTS ON UDA METHODS AND ON FINE-TUNING PERFORMANCE

This appendix presents a comparative analysis of THE UDA methods and their impact on the fine-tuning performance of our model. Figure 9 shows the ROC-AUC scores for different UDA methods as we increase the percentage of target domain data used for fine-tuning. We see two things: i) No configuration of the UDA methods improves the model's performance significantly, and ii) the robustly trained models also do not benefit from faster finetuning with fewer samples, as their performance when further finetuned with target-data samples is similar to the original model after finetuned on the same splits. In short, we believe that traditional UDA methods do not show promise for the NLI task.

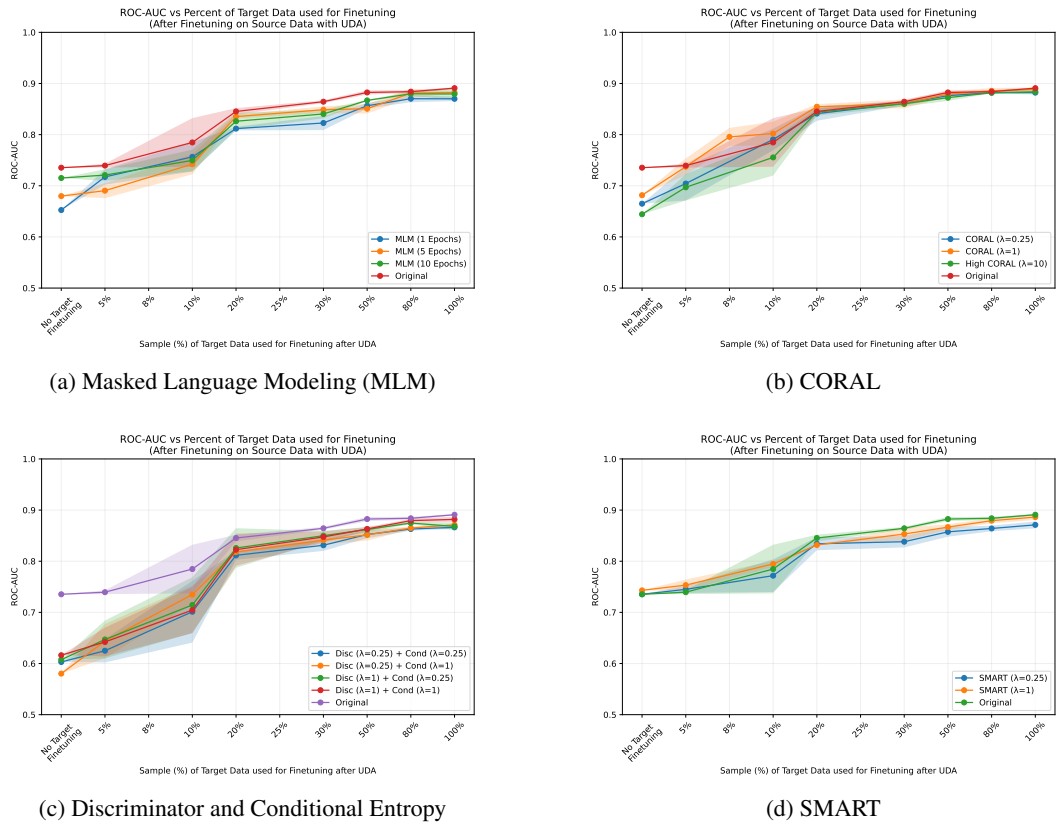

(a) Masked Language Modeling (MLM)

(b) CORAL

(c) Discriminator and Conditional Entropy

(d) SMART

Figure 9: Performance comparison of different UDA methods across fine-tuning percentages

## D.10   GENERATION TIME FOR AUTO-GDA

To provide a complete picture of our algorithm, we include runtimes that we obtained for each of the steps of Auto-GDA. We run Auto-GDA on the smallest and largest dataset, LFQA and RAGtruth-QA respectively, on our hardware (single Nvidia Tesla V100 GPU, 64GB RAM, 8 Intel Xeon CPUs) and provide individual runtimes in Table 16.

## E   PROMPTS

### E.1   INITIAL GENERATION PROMPTS

We use two prompts to generate intitial samples that differ according to the respective target labels. In practice we use a maximum of 4 few shot samples, or the number of samples available in the train dataset for a given evidence $e$.

| Dataset | Real / Generated Size | Initial Generation | Mutation | Selection | Fine-tuning | Total (1 iteration) |
|---|---|---|---|---|---|---|
| LFQA | 171 / 336 | 90.12 | 188.12 | 90.97 | 34.75 | 403.96 |
| RAGTruth-Summ | 3662 / 5032 | 650.38 | 1562.78 | 1161.11 | 432.84 | 3807.11 |

Table 16: Generation times (sec.) obtained when running Auto-GDA on the largest and smallest dataset on our hardware. The total time for one iteration of Auto-GDA is roughly 4x-6x more than generating few-shot data only. We would like to highlight that runtimes depend rather strongly on the latency of the API calls (and how much they are parallelized).

Positive (entailed) prompt:

> Human: You are given the following document wrapped in <document> </document> tags:
> <document>**DOCUMENT**</document> Your task is to generate summaries from a document. Here are some examples of how the summaries could look like:
>
> Note however that some of the samples contain incorrect information that is not part of the document! Here are the examples:
> <example 0>**EXAMPLE0**</example 0>
> <example 1>**EXAMPLE1**</example 1>
> Now your task is to generate **N** summaries from the document. However, unlike some of the examples given above, the summaries must be entirely supported by the document. Only include information that is directly inferrable from the document. It is also important that the summaries reflect the style, length and wording of examples. If there are common patterns or sentence structures in the examples summaries, the created summaries should reflect those. Each summary is identified with an integer from 0 to **N-1**. The summaries must be wrapped in <summary #></summary #> tags, where # is replaced with the summary id.
> Assistant:

To generate non-entailed samples, the following modified prompt is used:

> Human: You are given the following document wrapped in <document> </document> tags:
> <document>**EVIDENCE DOCUMENT**</document> Your task is to generate summaries from a document. Here are some examples of how the summaries could look like:
>
> Note however that some of the samples contain incorrect information that is not part of the document! Here are the examples:
> <example 0>**CLAIM EXAMPLE0**</example 0>
> <example 1>**CLAIM EXAMPLE1**</example 1>
> Your task is to generate **N** summaries from the document. However, now all of the summaries must contain at least one piece of non-factual information. This can be some information that is not present in the document or some information that is contradictory to the information in the document, but intuitively appears to make sense. Otherwise they reflect the style, length and wording of examples. If there are common patterns or sentence structures in the examples summaries, the created summaries should reflect those. Modify different pieces of information at different places in the document. Each summary is identified with an integer from 0 to **N-1**. The summaries must be wrapped in <summary #></summary #> tags, where # is replaced with the summary id. Assistant:

### E.2 ENTAILMENT PREDICTION PROMPT

We use the following prompt to compute entailments with the LLMs. It stems from Tang et al. (2022), however instead of answering Yes/No the LLM is prompted to anser with "0"/"1", which has the advantage that the token probabilities can be used to compute an uncertainty score.

> Determine whether the provided claim is consistent with the corresponding document. Consistency in this context implies that all information presented in the claim is substantiated by the document. If not, it should be considered inconsistent.
> Document: **EVIDENCE DOCUMENT**
> Claim: **CLAIM**
> Please assess the claim's consistency with the document by responding with either "1" (consistent) or "0" (inconsistent). Do not output anything else. Answer:

### E.3 LLM Partial Rephrasing Prompt

We use the following prompt to instruct the LLM to only rephrase specific parts of a sentence that are masked out with "_".

---

Your task is to fill in the gaps in a document indicated with "_" with additional details. If there is no gaps, please output the input text. The number of "_" indicates the approximate number of words that should be filled into each gap. While slight deviations (e.g., one word more or less) are permissible, the filled in text should respect the length indicated through the number of "_". **Do not change the text outside the gaps and do not include gaps in the final output.** You will generate **N** different completions of the document. Each completed document is identified with an integer from 0 to **N-1**. The document with the blanks filled must be wrapped in <answer #></answer #> tags, where # is replaced with the id of the filled-in document. You will now see the original document, but you will have to generate different versions that preserve the meaning by filling the gaps.

Here is the original: <document>**EVIDENCE DOCUMENT**</document>

The document including the gaps is: <document>**DOCUMENT WITH WORDS MASKED**</document>

Assistant:

---

