# OpenReview forum: "Auto-GDA: Automatic Domain Adaptation for Efficient Grounding Verification in Retrieval-Augmented Generation"
_ICLR.cc/2025/Conference — ICLR 2025 Poster_

### Official Review · Reviewer_4u6k · 2024-10-28

**Soundness:** 3
**Presentation:** 3
**Contribution:** 2
**Rating:** 6
**Confidence:** 4

**Summary:**

The paper considers the problem of domain adaptation of the natural language inference models (NLI) which are often used in retrieval-augmented generation (RAG) to judge the entailment of the generated response from the retrieved context. The paper proposes Auto-GDA, a method for generating synthetic data for a given domain, that can be used for finetuning an NLI model to improve its performance in this domain. The proposed method is iterative and consists of three steps: (1) seed data generation using an LLM; (2) data augmentation e.g. using paraphrasing or sentence deletion; and (3) data filtering, to minimize their proposed enhanced distribution matching objective; steps 2 and 3 can be repeated iteratively. The method is tested on several datasets that provide human labels for NLI, and compared versus existing off-the-shelf systems and several ablations.

**Strengths:**

* A relevant research direction of adapting RAG components to user domains
* Comprehensive related work section
* Detailed description of the proposed approach, used datasets, baselines, and experimental details
* The proposed method is compared to a series of existing NLI solutions on several datasets, and inference time is also compared for various methods

**Weaknesses:**

1. The presented Auto-GDA method is rather complex (involving multiple steps and components, including heuristic augmentation techniques) and has several important hyperparameters, such as $\lambda_d$ and $\lambda_u$ in eq. (2), population sizes M and K, or the number of iterations. Hyperparameter tuning requires running the proposed approach 50 times (line 446), including training of the NLI model on the generated data (from my understanding). At the same time, improvements over simply using LLM-generated data are quite modest (row 4 vs 2 in Table 2).
    - Furthermore, the high cost of running hyperparameter optimization, augmentation, and data selection in the proposed approach, motivates the substantial increase of data points in the simple baseline of using LLM-generated data, to make their computational costs similar. This would make the baseline stronger and reduce its difference versus the proposed approach even further. [Update during discussion: addressed in the authors' response]
    - Performance gains versus out-of-the-box NLI models are also rather small, e.g. comparing the best performing (domain-adapted) Auto-GDA versus the best performing “complex” method out-of-the-box (83.7 vs 80.5, 86.7 vs 85.4, 92.5 vs 90.4, 88.3 < 89.4), or Auto-GDA (Flan-T5) vs MiniCheck-T5 (75.6 \approx 75.4, 68.7 < 74.1, 82.4 vs 79.1; only one high improvement 78.3 vs 64.0).
2. I would expect more ablations of the proposed approach, e.g. testing the removal of each of the terms in eq. (2), or on the contrary using only one of these terms for filtering. [Update during discussion: the ablation was provided, showing that the proposed enhanced distribution matching objective seems to be mostly reducible to a single term]
3. The motivation for the proposed approach is to improve RAG pipelines in domain-specific scenarios, however no experiments with domain-specific RAG are presented to demonstrate these improvements. For example, domain-specific RAG scenarios considered in [1] could act as potential testbeds, e.g. RobustQA [2]. [Update during discussion: addressed in the authors' response]

[1] Dongyu Ru et al. RAGChecker: A Fine-grained Framework for Diagnosing Retrieval-Augmented Generation. NeurIPS 2024

[2] Rujun Han et al.  RobustQA: Benchmarking the Robustness of Domain Adaptation for Open-Domain Question Answering. Findings of ACL 2023

**Questions:**

1. What is the time of running each step in Auto-GDA (steps 1-2-3 and NLI model training), for different datasets (with their sizes)?
2. One of the arguments for the necessity of domain adaptation in the introduction is that “inputs may follow a specific format due to the RAG prompt template” (line 79). Why not pass evidence and claims to NLI models without this template, if it reduces domain shift and hence improves performance?
3. What criteria is used to select optimal hyperparameters using optuna? Is it performance on some dataset (which one)?
4. How do you tune hyperparameters of unsupervised domain adaptation baselines?
5. Are there particular reasons why Vectara-2.1 outperforms AlignScore on RAGTruth and vice versa on other datasets? E.g. due to some specific training data or algorithm specifics.
6. Do you plan to release open-source code for the proposed approach?
[Update during discussion: all questions answered]


Comments
* Due to the high amount of notations, it may be hard to follow the method sometimes, e.g. trying to remember what a particular notation means.
* Line 43: “even when the most capable LLMs are used with RAG, hallucination rates of 20 – 30% persist (Santhanam et al., 2021).”  too old reference
* Figure 1: “RAG task performance (ROC-AUC).” Unclear x label: it seems that it is RAG performance (unclear how measured with ROC-AUC), but it is NLI performance for RAG as far as I understand
* Line 233: “we also generate realistic initial samples using LLMs”. Use a more specific term than “samples”
* Line 384: “Optimizing the objective for a subset Q”. What does Q refer to here?
* Line 429: better define “complex” category

---

> ### Author Response · Authors · 2024-11-19
> **Response to Reviewer 4u6k**
>
> We thank the reviewer for the detailed review and the positive comments highlighting the relevance, descriptions, and comparative evaluation of our work. We will address the remaining points below.
>
> > [...] the high cost of running hyperparameter optimization, augmentation, and data selection in the proposed approach, motivates the substantial increase of data points in the simple baseline of using LLM-generated data. This would make the baseline stronger and reduce its difference versus the proposed approach even further
>
> Thank you for this point. Following the suggestion by the reviewer, we generate more LLM data using our few-shot prompting strategy and fine-tune the DeBERTa model on LLM-generated datasets in the range of 0.5x - 8x the size of the datasets used for comparison in the main paper.
>
> We obtain the following results (ROC-Scores of the DeBERTaV2 model fine-tuned on the data):
>
> | LLM dataset size ratio (to paper) | 0.5x           | 1x             | 2x             | 4x             | 8x             |
> |-----------------------------------|----------------|----------------|----------------|----------------|----------------|
> | LFQA                              | 0.916 +- 0.007 | 0.924 +- 0.006 | 0.903 +- 0.013 | 0.896 +- 0.007 | 0.868 +- 0.003 |
> | SummEdits                         | 0.880 +- 0.003 | 0.876 +- 0.002 | 0.876 +- 0.005 | 0.868 +- 0.002 | 0.855 +- 0.005 |
> | RAGTruth-Summ                     | 0.821 +- 0.003 | 0.808 +- 0.005 | 0.791 +- 0.007 | 0.797 +- 0.001 | 0.783 +- 0.004 |
> | RAGTruth-QA                       | 0.808 +- 0.007 | 0.828 +- 0.006 | 0.830 +- 0.006 | 0.834 +- 0.004 | 0.821 +- 0.005 |
> | Average                           | 0.860          | 0.856          | 0.849          | 0.844          | 0.841          |
>
> Perhaps surprisingly, increasing the number of generated samples does not increase the performance of fine-tuned models. We attribute this to the observation that (even when setting a higher temperature), LLMs will generate highly similar samples after a certain dataset size is reached, not further improving performance of a fine-tuned model. Performance even decreases again, potentially due to overfitting on the synthetic data.  We therefore conclude that even with more LLM data, the performance does not reach the level of Auto-GDA. We have included results and this discussion in Appendix D.6.
>
> > Performance gains versus out-of-the-box NLI models are also rather small, e.g. comparing the best performing (domain-adapted) Auto-GDA versus the best performing “complex” method out-of-the-box
>
> Besides the absolute gains, we would like to stress that our models have several other advantages: First, they do not require chunking longer text into pieces as done in AlignScore. This proceduce is prone to breaking necessary context. Second, we observe that inference time for our models is **2.7x faster** than AlignScore on average, which is a substantial improvement. When considering solely performance, we therefore think the non-finetuned models should serve as a reference as they have identical other characteristics. Here, we observe a consistent improvement over datasets and NLI model architectures **improving by over 10 points** on average.
>
> Depending on the application, we are further convinced that the gains reported can be considered a distinct improvement. Nominally small gains are common in the NLI literature: MiniCheck reports an average 2.2-4.3 points of improvement over AlignScore (Table 2 in Tang et al.), while Alignscore reports an average 3.8 points over next best UniEval (Table 2 in Zha et al.). This particularly holds when performance approaches the supposed upper bound of SotA models trained on human-labeled data (see Table 2 in the submission).

---

> ### Author Response · Authors · 2024-11-19
> **Response to Reviewer 4u6k (cont.)**
>
> > 2. I would expect more ablations of the proposed approach, e.g. testing the removal of each of the terms in eq. (2), or on the contrary using only one of these terms for filtering
>
> Thank you for this point about the terms in the selection criterion. Following the suggestion by the reviewer, we ran an additional experiment where we only use one of the components of the loss in Eqn. 2 at a time. We perform 5 runs for LFQA and SummEdits and provide preliminary results for a single run on the RAGTruth parts due to the time constraints of the rebuttal. Our experiment  leads to the following final AUC-ROC-Scores after fine-tuning a DeBERTaV2 model:
>
> | Dataset        | All-Terms        | Only Distance  | Only Label-Cert. | Only Utility    |
> |----------------|------------------|----------------|------------------|-----------------|
> | LFQA           | 0.925 +- 0.004   | 0.916 +- 0.004 | 0.919 +- 0.003   | 0.917 +- 0.005  |
> | Summedits      | 0.883 +- 0.002   | 0.877 +- 0.004 | 0.866 +- 0.002   | 0.881 +- 0.002  |
> | RAGTruth-Summ  | 0.861            | 0.805          | 0.860            | 0.587           |
> | RAGTruth-QA    | 0.827            | 0.756          | 0.814            | 0.823           |
>
> Despite the differences being rather small for some combinations, all terms contribute to the final performance.
> We would further like to stress that our hyperparameter search grid includes zero values for $\lambda_u$ and $\lambda_d$. If the terms had no positive contribution to the loss, the optimizer would have likely set them to zero or to very small values. However, we see that for the best performing models (upper part of Table 6) only $\lambda_d$ for RAGTruth-QA is set to a small value providing further evidence that the other terms are indeed necessary for optimal performance on most datasets. We have added this discussion to Appendix D.7.
>
> > 3. The motivation for the proposed approach is to improve RAG pipelines in domain-specific scenarios, however no experiments with domain-specific RAG are presented [...] domain-specific RAG scenarios considered in [1] could act as potential testbeds, e.g. RobustQA [2].
>
> We focus on the problem of grounding verification in RAG systems. We follow the standard set by other papers in the literature, e.g., MiniCheck (Tang et al., EMNLP 2024), that consider the specific subproblem of fact-checking in RAG. Evaluating end-to-end RAG systems requires separate analysis on the retrieval and generation components, which is not within the scope of this paper. We thank the reviewer for the RAGChecker reference and have added it to the outlook section.
>
> We are also aware of the RobustQA dataset mentioned. Unfortunately, the dataset does not provide LLM answers with human labels for groundedness, hence we were not able to use this dataset in our evaluation.
>
>
> > Questions:
> > 1. What is the time of running each step in Auto-GDA
>
> We rerun Auto-GDA with the smallest and largest dataset, LFQA and RAGtruth-QA respectively, on our hardware (single Nvidia Tesla V100 GPU, 64GB RAM, 8 Intel Xeon CPUs) and provide individual runtimes below:
>
> | Dataset                        | LFQA      | Ragtruth-QA |
> |--------------------------------|-----------|-------------|
> | Original Size / Generated Size | 171 / 336 | 3662 / 5032 |
> | Initial Generation Time (sec)  | 90.12     | 650.38      |
> | Mutation-Time (sec)            | 188.12    | 1562.78     |
> | Selection-Time (sec)           | 90.97     | 1161.11     |
> | Finetuning-Time (sec)                  | 34.75     | 432.84      |
> | Total Time (sec)               | 403.96    | 3807.11     |
>
> As expected the mutation step takes up most time, followed by the selection step (which includes forward passes of all samples to assess their utility). Overall, an iteration of Auto-GDA takes roughly 4-6x more time than just plain generation (making the additional LLM generated data in the experiment above comparable to Auto-GDA in terms of runtime).
>
> The additional generation time however results in substantially reduced inference time (up to 90% inference time reduction compared to LLM calls). For real-time constrained applications this gain can be crucial.
>
> > 2. Why not pass evidence and claims to NLI models without this template, if it reduces domain shift [...]
>
> Thank you for this question. For some applications, it might be hard to remove the prompt template without breaking the functionality as the template may contain relevant information for processing. For instance, Chiang et al. (2024) include additional meta-data as document age and source in RAG. The template may also include information from the system designer (e.g., system prompts). Fully omitting the template would result in unstructured information that is hard to make sense of for the LLM, potentially resulting in a performance degradation as well.
>
> Nevertheless, we believe that the format (e.g., writing, language, style) and domains of the documents in the knowledge base are most responsible for domain shift.

---

> ### Author Response · Authors · 2024-11-19
> **Response to Reviewer 4u6k (cont.)**
>
> > 3. what criteria is used to select optimal hyperparameters using optuna?
>
> Thanks for this relevant question. We use a tiny holdout dataset of 30 - 150 samples (as mentioned in l.183 - l.186) to assess performance of a fine-tuned DeBERTaV2 model. The ROC score of the fine-tuned model is used for hyperparameter selection. We made sure that fine-tuning on this data straight away does not yield the performance of Auto-GDA (Table 12). We have pointed this out in Section 5 our revised manuscript.
>
> > 4. How do you tune hyperparameters of unsupervised domain adaptation baselines?
>
> In general, we use the same training hyperparameters (epochs, lr, ...) between the original models and the UDA-augmented models. As stated at the end of Appendix C.6. we set the coefficient of the respective regularization terms to 0.5, after running hyperparameter optimization with a coarse grid search.
> Overall, we experimented with multiple values of the UDA regularization penalty parameter which led to no significant changes. Some results for different values of lambda with limited effect can be found in Figure 9 (Appendix) of our paper. This finding is in line with DIRT-T paper (Shu et al., 2018, Appendix B) which also argues that the approaches are quite invariant to hyperparameter choices.
>
> > 5. Are there particular reasons why Vectara-2.1 outperforms AlignScore on RAGTruth and vice versa on other datasets? E.g. due to some specific training data or algorithm specifics.
>
> Unfortunately, although the trained Vectara-2.1. model is OpenSource, the training details of Vectara-2.1. were not publicly released. However, the authors use RAGTruth in their evaluation, so it seems plausible  that they put in some effort to specifically cover datasets of similar characteristics.
>
> > 6. Do you plan to release open-source code for the proposed approach?
>
> We commit to fully open-sourcing our code in case of acceptance. A preliminary version of the code is already publicly visible along with this submission (supplementary material). We have added a corresponding statement to Appendix C.1.
>
> > Comments
>
> > hallucination rates of 20 – 30% persist (Santhanam et al., 2021).” too old reference
>
> We have replaced the reference by Niu et al. (2024) who observe more than one hallucinated text span per 100 output tokens on Llama-2 models and Chen et al. (2023) show hallucination rates of 15% to over 30% even for GPT-3.5.
>
> We are grateful for the reviewer’s other minor comments regarding the clarity of the write-up and have done our best to address them in our revised manuscript. We are open to further suggestions on how to improve our notation. Let us know if there are any other passages that can be made more precise.
>
> ---------------
>
> We thank the reviewer again for the detailed review and for the relevant questions which allowed us further clarify the description of our approach. We sincerely hope that we have been able to address the main points regarding the sample size for the LLM-data baseline, the ablation study, generation time, and hyperparameters with the revisions and our additional experimentation. In that case, we kindly ask the reviewer to reconsider their overall assessment in light of this rebuttal, the general comment, and the other reviews. We are also happy to answer any further questions.
>
> ------------------
>
> **References**
>
> Chiang, Cheng-Han, and Hung-yi Lee. "Do Metadata and Appearance of the Retrieved Webpages Affect LLM’s Reasoning in Retrieval-Augmented Generation?." Proceedings of the 7th BlackboxNLP Workshop: Analyzing and Interpreting Neural Networks for NLP. 2024.
>
> Niu, C., Wu, Y., Zhu, J., Xu, S., Shum, K., Zhong, R., ... & Zhang, T. RAGTruth: A hallucination corpus for developing trustworthy retrieval-augmented language models. arXiv preprint arXiv:2401.00396, 2024
>
> Rui Shu, Hung H Bui, Hirokazu Narui, and Stefano Ermon. A dirt-t approach to unsupervised
> domain adaptation. ICLR 2018
>
> Liyan Tang, Philippe Laban, and Greg Durrett.. MiniCheck: Efficient Fact-Checking of LLMs on Grounding Documents. In Proceedings of the 2024 Conference on Empirical Methods in Natural Language Processing, pages 8818–8847, Miami, Florida, USA. Association for Computational Linguistics, 2024
>
> Hung-Ting Chen, Fangyuan Xu, Shane A Arora, and Eunsol Choi. Understanding retrieval augmentation for long-form question answering. arXiv preprint arXiv:2310.12150, 2023
>
> Yuheng Zha, Yichi Yang, Ruichen Li, and Zhiting Hu. AlignScore: Evaluating factual consistency with a unified alignment function. Proceedings of the 61st Annual Meeting of the Association for Computational Linguistics (Volume 1: Long Papers), pp. 11328–11348, 2023

---

> > ### Comment · Reviewer_4u6k · 2024-11-22
> > **Response to the authors**
> >
> > Thank you for the detailed reply to my comments and providing additional results, as well as updating the pdf!
> >
> > __An experiment with 0.5x-8x LLM-generated data__
> >
> > Thank you for providing these results, they are very insightful! This analysis addresses my Weakness 1.1. I would suggest including the discussion you provided about your hypotheses for these results in the same Appendix.
> >
> > __Performance gains__
> >
> > > When considering solely performance, we therefore think the non-finetuned models should serve as a reference as they have identical other characteristics.
> >
> > Could you please clarify the argumentation for this sentence, as I did not understand it.
> >
> > > Nominally small gains are common in the NLI literature
> >
> > I agree that this can be the case, but I am also considering improvements in the context of the effort needed to achieve them, i.e. running the complex multi-iteration data augmentation procedure with hyperparameter optimization. Reviewer Wxqg mentions similar concerns. The references MiniCheck and AlignScore are much easier to use, i.e. out-of-the-box, from my understanding.
> >
> > __Ablations__
> >
> > Thank you for providing these updated results, they are very insightful! Did you perform any analysis why using only the Utility term leads to so low results on RAGTruth Summary, in contrast to all other datasets where this term is basically enough to be used alone?
> >
> > > the optimizer would have likely set them to zero or to very small values
> >
> > I am not sure we have evidence this is fully true, since the hyperparameters optimizer probably does not perform full grid search of all possible hyperparameter combinations.
> >
> > __Other datasets such as Robust QA__
> > > Robust QA does not provide LLM answers with human labels for groundness
> >
> > The author's comment resolves my concern about other datasets (Weakness 3).
> >
> > __Time of running each step in Auto-GDA__
> >
> > Thank you for providing these results. I suggest including these measurements in Appendix and pointing to them in the main text.
> >
> > __To summarize,__ I increase the Soundness score and the Overall rating, based on the provided new results.
> >
> > However, I prefer to keep my rating for Contribution, because essentially the work shows advantages of using data augmentation, which is a well-known technique to improve domain performance, while the proposed enhanced distribution matching objective seems to be mostly reducible to a single term in the new ablation results.
> >
> > I also agree with the concern raised by two other reviewers, than while the work proposes to optimize NLI to a given data distribution, in practice it could happen that this optimized NLI model would accidentally face inputs outside of this distribution, and it is unclear how this model would handle such examples. At the same time, this would not be a problem for generic models such as MiniCheck.

---

> ### Author Response · Authors · 2024-11-24
> **Response to Follow-Up by Reviewer 4u6k**
>
> Thank you for engaging with our rebuttal! We appreciate your consideration and your constructive feedback. We address the follow-up questions below and have incorporated our provided responses in the paper draft where applicable.
>
> > An experiment with 0.5x-8x LLM-generated data: This analysis addresses my Weakness 1.1. I would suggest including the discussion you provided about your hypotheses for these results in the same Appendix.
>
> Thank you, we are happy to hear that this point could be resolved. We have added the discussion points to the Appendix D.6. of our paper as suggested.
>
> > Performance gains: "When considering solely performance, we therefore think the non-finetuned models should serve as a reference as they have identical other characteristics." Could you please clarify the argumentation for this sentence, as I did not understand it.
>
> > [...] The references MiniCheck and AlignScore are much easier to use, i.e. out-of-the-box, from my understanding.
>
> In this statement, we wanted to point out that each model has different strengths and weaknesses. These include (amongst others) the model’s predictive performance, inference latency, and whether sentences are chunked, potentially breaking context in multi-sentence inputs. Depending on the constraints of the application in mind, different requirements are in place and determine maximum achievable performance. We therefore think the most insightful comparison is between the Auto-GDA finetuned and non-finetuned base models, as they possess identical other characteristics. We include the other models in Tab. 1 to provide an explicit characterization of these trade-offs, but would like to point out that this represents only one dimension of a model's full performance characteristics, which are interdependent. Our models perform very well in terms of both inference latency and predictive performance.
>
> We agree that AlignScore and MiniCheck have their own desirable characteristics and use-cases. As we strive to give a most objective discussion of the pros and cons of competing approaches and to make actionable recommendations for practitioners, we have included a clarifying remark in this regard in the discussion section (Section 6). We thank the reviewer for raising this point.
>
> >  Did you perform any analysis why using only the Utility term leads to so low results on RAGTruth Summary, in contrast to all other datasets
>
> Unfortunately, we do not have a definitive answer to this question. We observe that RAGTruth-Summary is likely the most difficult dataset for initial data generation due to its length and complexity of the evidence examples. We have the following average number of tokens in the evidence of the processed datasets:
>
> * RAGTruth-QA:  381.95
> * RAGTruth-Summary: 546.14
> * SummEdits: 438.42
> * LFQA: 377.31
>
> Evidences from RAGTruth-Summary have more than 100 tokens more on average than those from all other datasets. This may be one reason for why we find that there is a non-negligible number of mislabeled and incoherent generations initially. For Auto-GDA to result in good data quality, the initial sample population must be filtered by either the label certainty or by embedding distance (not as effective) to rule out a substantial part of these low-quality samples. Utility does not measure sample quality - on the contrary, mislabeled samples have high loss and therefore high utility. This is why we believe utility alone as a criterion rather hurts performance in face of low-quality initial samples as for RAGTruth-Summary.
>
> > I am not sure we have evidence this is fully true, since the hyperparameters optimizer probably does not perform full grid search of all possible hyperparameter combinations.
>
> Yes, the reviewer is right that there is no technical guarantee. However, there are only 3 parameters and we perform 50 trials, so we observe the hyperparameter space to be covered reasonably dense. The trials further focus on areas in the parameter space with best performance, leading us to believe that these combinations would indeed have been discovered if they resulted in best performance.
>
> > Time of running each step in Auto-GDA Thank you for providing these results. I suggest including these measurements in Appendix and pointing to them in the main text.
>
> Thank you for this suggestion. We have done so in our revised manuscript: Results can be found in Appendix D.10  / Table 16 with a reference added in Section 5.3.
>
> -----------------------
>
> We sincerely thank the reviewer again for responding and are happy to answer any further questions.

---

> > ### Comment · Reviewer_4u6k · 2024-11-28
> >
> > Thank you for your answers and additional comments!

---

### Official Review · Reviewer_ykaR · 2024-11-03

**Soundness:** 3
**Presentation:** 3
**Contribution:** 3
**Rating:** 6
**Confidence:** 3

**Summary:**

This paper tackles the issue of hallucinations in Large Language Models (LLMs) used in retrieval augmented generation (RAG) applications, where verification of generated outputs through natural language inference (NLI) models is essential. The authors propose Automatic Generative Domain Adaptation (Auto-GDA), an unsupervised framework that generates high-quality synthetic data to fine-tune lightweight NLI models for specific RAG domains. Key contributions include formalizing the unsupervised domain adaptation problem, developing the Auto-GDA framework for efficient sample selection, and demonstrating that models fine-tuned with Auto-GDA outperform weak teacher models and approach the performance of human-labeled references, all while achieving significantly lower latency than LLMs. This work presents a promising solution to enhance NLI model performance in real-time RAG applications.

**Strengths:**

- The proposed Automatic Generative Domain Adaptation (Auto-GDA) offers a novel approach to unsupervised domain adaptation, effectively generating high-quality synthetic data to fine-tune NLI models tailored for specific RAG contexts.
- Empirical results demonstrate that models fine-tuned with Auto-GDA significantly outperform weak teacher models and achieve performance levels comparable to those using human-labeled data, indicating its effectiveness in improving NLI model accuracy.
- The paper is well written and easy to understand.

**Weaknesses:**

- The effectiveness of the Auto-GDA framework relies heavily on the quality of the synthetic data generated. Poorly generated data could lead to suboptimal fine-tuning and negatively impact NLI model performance.
- Although the framework aims to address domain mismatches, (in my own opinion) there may still be challenges in generalizing to highly diverse or previously unseen domains, potentially reducing the model's effectiveness in broader applications. I'm curious how well the proposed method performs under such extreme conditions?

**Questions:**

Refer to weaknesses.

---

> ### Author Response · Authors · 2024-11-19
> **Response to Reviewer ykaR**
>
> Thank you for your thoughtful review and for recognizing the strengths of our work, including its ability to adapt to specific RAG contexts and our supportive evaluation results. We will address your specific points and questions raised below.
>
> > [...] Poorly generated data could lead to suboptimal fine-tuning and negatively impact NLI model performance.
>
> We agree that the initial data plays a crucial role in Auto-GDA. However, we specifically designed our framework to be able to deal with subpar samples in the initially generated data. In general, we do not rely on a single model (e.g., the initial data generator) but combine discriminative and generative models throughout the generation loop to make the algorithm more fault-tolerant. We specifically craft the objective for the selection step to filter out low-quality samples, e.g., incorrect labels or are unrealistic generations. For instance, data with seemingly incorrect labels should be assigned a low $r_0$ and atypical data have high embedding distance from the target samples effectively reducing the chance of the data and augmented versions thereof being selected for the final fine-tuning.
>
> Inspired by the reviewers’ suggestion, we are interested in testing the effectiveness of this design in practice. Mislabeled samples in the fine-tuning dataset can severely affect performance of a fine-tuned model (Wang et al., 2023). To test the robustness of Auto-GDA, we manually flip 50% of the labels in the initial data to study its effect and trace these samples through the generation process. We use the LFQA-Verification dataset and perform 5 runs with different initial generation seeds (using the same hyperparameters as in Table 1 otherwise). First, while 50% of the data have flipped labels initially, in the data selected after one iteration of Auto-GDA, only **10.0% +- 1.1%** are the initial flipped data or augmentation of the data with the flipped labels. This highlights that our selection step manages to filter out most of the mislabeled samples. After fine-tuning on the generated data, the ROC-AUC score drops slightly by 1.1 points:
>
> |                               | Original Performance | With 50% flipped initial labels |
> |-------------------------------|----------------------|---------------------------------|
> | ROC-AUC of fine-tuned DeBERTa |  0.925 ± 0.009       |  0.914 ± 0.007                  |
>
> Despite the substantial amount of mislabeled initial data (50%), most of these samples do not make it through the selection step and the drop in performance remains small, highlighting that Auto-GDA is quite robust to initial data quality issues. We include this finding in Appendix D.4.
>
> >  (in my own opinion) there may still be challenges in generalizing to highly diverse or previously unseen domains, potentially reducing the model's effectiveness in broader applications. I'm curious how well the proposed method performs under such extreme conditions?
>
> We specifically select datasets with open-ended (LFQA, contains general queries for explanations of different phenomena) and diverse domains (SummEdits, contains 10 subdomains) in our evaluation to provide a realistic performance assessment. While in principle, our approach works for any unseen domain, it requires access to the corpus of such domains.  Without any access to a corpus, our approach would not work, i.e., it does not tackle generalization to completely unseen inputs. In such scenarios, practitioners should do their best to find the most representative corpus to the target domain. We therefore see our framework as a promising approach for non-standard domains, where documents and queries are available. This is the case in RAG, where the system designer has access to the document base. We have clarified the scope in our revised manuscript (limitation section).
>
> ---------------
>
> We thank the reviewer again for the valuable feedback and hope to have adressed the remaining points regarding the robustness to data quality issues in the initial data and the scope of our method. Let us know if you have any further questions.
>
> ---------------------
> **References**
>
> Wang, S., Tan, Z., Guo, R., & Li, J. (2023, December). Noise-Robust Fine-Tuning of Pretrained Language Models via External Guidance. In Findings of the Association for Computational Linguistics: EMNLP 2023 (pp. 12528-12540).

---

### Official Review · Reviewer_Wxqg · 2024-11-06

**Soundness:** 3
**Presentation:** 3
**Contribution:** 3
**Rating:** 8
**Confidence:** 2

**Summary:**

This paper introduces a novel framework to enhance the performance of NLI models in verifying retrieved evidence within retrieval-augmented generation settings. The paper addresses the issue of performance drop in out-of-domain inputs. To alleviate this problem, the authors propose an automatic generative domain adaptation method to fine-tune NLI models in an unsupervised manner. In the proposed method, this framework considers both diversity and quality by using a sequential augmentation technique and optimizing a distribution-matching objective in the data generation process. Experimental results demonstrate that the NLI model fine-tuned with the proposed method achieves performance closer to that of LLM models without sacrificing efficiency.

**Strengths:**

- The proposed method is novel and practical in RAG scenarios.
- This manuscript is clearly written and easy to follow.
- The experimental results are well conducted, showing advantages in terms of accuracy and efficiency.

**Weaknesses:**

- The paper would benefit from a more detailed analysis to clearly demonstrate the robustness of the proposed method across various domains. In real-world scenarios, domain boundaries are often ambiguous, and it’s common to encounter mixed or overlapping domain data. By evaluating the model in a multi-domain or domain-mixing setting, the authors could provide stronger evidence of its robustness and practical applicability in complex, realistic cases.

- It would be valuable to include an analysis of how the model performance depends on the quality of the initial synthetic data generation. If initial data is inaccurately generated, it might negatively influence the model’s performance. An examination of whether the model can correct or adapt to potential errors in this initial phase would clarify the method’s resilience to suboptimal synthetic data.

- The paper’s selective objective function appears complex, suggesting that the model could be highly sensitive to hyperparameter choices. Observing large variations in hyperparameter values for each dataset implies that tuning these parameters may be challenging. Providing further insights into the model’s hyperparameter sensitivity and offering guidelines for tuning could improve the approach's usability and reliability in diverse settings.

**Questions:**

See the weaknesses

---

> ### Author Response · Authors · 2024-11-19
> **Response to Reviewer Wxqg**
>
> We thank the reviewer for their positive review highlighting the novelty, writing, and experimental evaluation of our work. We will address the remaining points individually below.
>
> > [...] By evaluating the model in a multi-domain or domain-mixing setting, the authors could provide stronger evidence of its robustness and practical applicability in complex, realistic cases.
>
> Thank you for this point. We agree that domain boundaries might not be well-defined and domains can be mixed in practice. We therefore made sure to choose realistic datasets in our evaluation, which have open-ended and mixed domains. The point of mixed domains is covered in the SummEdits dataset, which consists of data from 10 different domains (such as news, law, financial documents, and more). For the LFQA-dataset, there is no sharp domain boundary either, the domain consists of questions from the “Explain me Like I’m 5”-dataset (ELI5) which covers many different topics and has no “sharp” boundaries. The performance of our approach on these datasets suggests that Auto-GDA also works well with open-ended domains and in the case where several domains are blended together. In general, if the evidence and claims used as seed data are representative of the domain, we expect our algorithm to work as described.
>
> > [...] An examination of whether the model can correct or adapt to potential errors in this initial phase would clarify the method’s resilience to suboptimal synthetic data.
>
> We agree that the initial data plays an essential role in Auto-GDA.  However, we designed our algorithm to be as fault-tolerant as possible and to be able to cope with some low-quality samples in the initial population. For instance, we do not solely rely on generative models, but use discriminative teacher models in the generation loop as well. The selection objective is specifically designed to filter out low-quality samples.
>
> Inspired by the reviewers’ suggestion, we are interested in testing the effectiveness of this design in practice. Mislabeled samples in the fine-tuning dataset can severely affect performance of a fine-tuned model (Wang et al., 2023). To test the robustness of Auto-GDA, we manually flip 50% of the labels in the initial data to study its effect and trace these samples through the generation process. We use the LFQA-Verification dataset and perform 5 runs with different initial generation seeds (using the same hyperparameters as in Table 1 otherwise).
>
> First, while 50% of the data have flipped labels initially, in the data selected after one iteration of Auto-GDA, only **10.0% +- 1.1%** are copies or augmentations of the data with the flipped labels.  After fine-tuning on the generated data, the ROC-AUC score drops slightly by 1.1 points:
>
> |                               | Original Performance | With 50% flipped initial labels |
> |-------------------------------|----------------------|---------------------------------|
> | ROC-AUC of fine-tuned DeBERTa |  0.925 ± 0.009       |  0.914 ± 0.007                  |
>
>
> We conclude that despite the substantial amount of mislabeled data (50%) the drop in performance remains small, highlighting that Auto-GDA is quite robust to data quality issues due to its fault-tolerant design. We include this experiment in Appendix D.4. of our revised manuscript.
>
> > [...] Providing further insights into the model’s hyperparameter sensitivity and offering guidelines for tuning could improve the approach's usability and reliability in diverse settings.
>
> Thank you for this suggestion. We perform hyperparameter search to have a principled procedure to choose values of the hyperparameters, but don’t think exhaustive hyperparameter optimization is essential for our algorithm. We identify two pieces of evidence for this assertion: (1) Our validation set for hyperparameter tuning is very small, in the range of 100 samples. This results in the selected hyperparameters already being quite noisy and potentially suboptimal. (2) We reexamine results from our hyperparameter optimization on the largest RAGTruth-QA dataset to assess sensitivity and to derive reasonable ranges. We find that when selecting both $\lambda_d>20$,  $\lambda_u>20$ and $0.3 <  \frac{\lambda_d}{\lambda_u} < 3$, all trials including the optimal one are within 2.2 ROC points with similar observations for other datasets.  As a general guideline, we therefore think setting $\lambda_d=\lambda_u$ to a value between 20 and 50 is a good choice. This is in line with our new ablation study (see response to reviewer ``4u6k’’) showing that each term’s contribution in the selection objective is important. Setting the parameters in this range ensures that all terms are of similar magnitude and contribute to the selection. In conclusion, choosing hyperparameters in a sensible range is important, but we don’t think that the algorithm's performance critically depends on the exact value of the parameters. We have added the guideline to the paper (Section 5).

---

> > ### Author Response · Authors · 2024-11-19
> > **Response to Reviewer Wxqg (cont.)**
> >
> > We thank the reviewer again for their positive review and sincerely hope to have addressed the questions regarding robustness to initial data and hyperparameter selection. Let us know if you have any further points you would like to discuss.
> >
> > ---------------
> > **References**
> >
> > Wang, S., Tan, Z., Guo, R., & Li, J. (2023, December). Noise-Robust Fine-Tuning of Pretrained Language Models via External Guidance. In Findings of the Association for Computational Linguistics: EMNLP 2023 (pp. 12528-12540).

---

### Author Response · Authors · 2024-11-19
**Summary of Revisions**

We thank all reviewers for their valuable comments which allowed us to clarify several points in our manuscript and to strengthen our evaluation. Most notably, we make the following changes:

* **Robustness to low-quality initial data:** We include a robustness study in response to reviewers ``Wxqg`` and ``ykaR`` highlighting that our selection objective is very capable of recovering from low-quality samples in the initial data (added to Appendix D.4.)
* **Effect of dataset size on LLM-data baseline:** In response to reviewer ``4u6k``, we investigate the effect of dataset size on the baseline of using plain LLM-generated data. We find that fine-tuning our model on additional data does not increase performance, but rather leads to a performance degradation, potentially due to overfitting (added to Appendix D.6.)
* **Ablation on terms in selection:** Following an additional suggestion by reviewer ``4u6k``, we run an ablation on the terms in our selection criterion, highlighting their contribution to the final performance. This finding is backed by results from the hyperparameter optimization which assigns non-zero weight to the corresponding weights of the terms in the objective (added to Appendix D.7.)
* **Guidelines for hyperparameters:** We clarify our hyperparameter tuning strategy in Section 5 and provide guidelines for choosing reasonable values without full optimization, addressing comments by reviewers ``Wxqg`` and ``4u6k``.
* **Scope and future work:** We point out the scope of our work (domains for which unlabeled examples are available) and add a remark that an evaluation including the complete RAG pipeline is out of the scope for this paper, but may complement our work in the future (Section 6)

We would also like to sincerely thank the reviewers for all other comments and have revised our manuscript accordingly. We highlighted all revised sections in red color. We hope that the points raised have been addressed and are glad to discuss any further questions.

---

### Meta-Review · Area_Chair_V3Gq · 2024-12-20

**Metareview:**

This submission aims to address the hallucination problem in RAG systems by enhancing a lightweight NLI model to verify whether the generated outputs are grounded in the retrieved documents. The authors provided a comprehensive rebuttal, effectively addressing most of the reviewers' concerns. Consequently, all reviewers are in agreement on the acceptance of this paper.

However, two major concerns remain after rebuttal:

- Limited technical contribution: The work essentially benefits from the usage of data augmentation, which is a well-known technique for improving domain transfer performance. Also related to this, one reviewer noted that the proposed objective can largely be reduced to a single term; and the approach involves tuning several hyperparameters. This limits the technical novelty of the contribution.

- Generaliability across domains: Since the proposed approach optimizes the NLI model for a specific data distribution, its generalizability to highly diverse or previously unseen domains may be limited. This constraint could impact its effectiveness in broader applications.

Considering the overall contributions and the addressed concerns, we recommend accepting this submission as a poster presentation.

**Additional Comments On Reviewer Discussion:**

The rebuttal has addressed most of the raised points. The unsolved points have been listed in the meta-review.

---

### Decision · Program_Chairs · 2025-01-22

Accept (Poster)